# Attribution of growing season evapotranspiration variability considering snowmelt and vegetation changes in the arid alpine basins

Tingting Ning[abc], Zhi Li[d], Qi Feng[ac]* , Zongxing Li[ac] and Yanyan Qin[b]

[a]*Key Laboratory of Ecohydrology of Inland River Basin, Northwest Institute of Eco-Environment and Resources,*

*Chinese Academy of Sciences, Lanzhou, 730000, China*

[b]*Key Laboratory of Land Surface Process and Climate Change in Cold and Arid Regions, Chinese Academy of*

*Sciences, Lanzhou 730000, China*

[c]*Qilian Mountains Eco-environment Research Center in Gansu Province, Lanzhou, 730000, China*

[d]*College of Natural Resources and Environment, Northwest A&F University, Yangling, Shaanxi, 712100, China*

* Correspondence to: Qi Feng (qifeng@lzb.ac.cn )

**Abstract:** Previous studies have successfully applied variance decomposition frameworks based on the Budyko equations to determine the relative contribution of variability in precipitation, potential evapotranspiration ($E_0$), and total water storage changes ($\Delta S$) to evapotranspiration variance ($\sigma_{ET}^2$) on different time-scales; however, the effects of snowmelt ($Q_m$) and vegetation ($M$) changes have not been incorporated into this framework in snow-dependent basins. Taking the arid alpine basins in the Qilian Mountains in northwest China as the study area, we extended the Budyko framework to decompose the growing season $\sigma_{ET}^2$ into the temporal variance and covariance of rainfall ($R$), $E_0$, $\Delta S$, $Q_m$, and $M$. The results indicate that the incorporation of $Q_m$ could improve the performance of the Budyko framework on a monthly scale; $\sigma_{ET}^2$ was primarily controlled by the $R$ variance with a mean contribution of 63%, followed by the coupled $R$ and $M$ (24.3%) and then the coupled $R$ and $E_0$ (14.1%). The effects of $M$ variance or $Q_m$ variance cannot be ignored because they contribute to 4.3% and 1.8% of $\sigma_{ET}^2$, respectively. By contrast, the interaction of some coupled factors adversely affected $\sigma_{ET}^2$, and the 'out-of-phase' seasonality between $R$ and $Q_m$ had the largest effect ($-7.6\%$). Our methodology and these findings are helpful for quantitatively assessing and understanding hydrological responses to climate and vegetation changes in snow-dependent regions on a finer time-scale.

**Keywords**: evapotranspiration variability; snowmelt; vegetation; attribution

## 1 Introduction

Actual evapotranspiration ($ET$) drives energy and water exchanges among the hydrosphere, atmosphere, and biosphere (Wang et al., 2007). The temporal variability in $ET$ is, thus, the combined effect of multiple factors interacting across the soil–vegetation–atmosphere interface (Katul et al., 2012; Xu and Singh, 2005). Investigating the mechanism behind $ET$ variability is also fundamental for understanding hydrological processes. The basin-scale $ET$ variability has been widely investigated with the Budyko framework (Budyko, 1961, 1974); however, most studies are conducted on long-term or inter-annual scales and cannot interpret the short-term $ET$ variability (e.g. monthly scales).

Short-term $ET$ and runoff ($Q_r$) variance have been investigated recently for their dominant driving factors (Feng et al., 2020; Liu et al., 2019; Wu et al., 2017; Ye et al., 2016; Zeng and Cai, 2015; Zeng and Cai, 2016; Zhang et al., 2016a); to this end, an overall framework was presented by Zeng and Cai (2015) and Liu et al. (2019). Zeng and Cai (2015) decomposed the intra-annual $ET$ variance into the variance/covariance of precipitation ($P$), potential evapotranspiration ($E_0$), and water storage change ($\Delta S$) under the Budyko framework based on the work of Koster and Suarez (1999). Subsequently, Liu et al. (2019) proposed a new framework to identify the driving factors of global $Q_r$ variance by considering the temporal variance of $P$, $E_0$, $\Delta S$, and other factors such as the climate seasonality, land cover, and human impact. Although

the proposed framework performs well for the *ET* variance decomposition, further
research is necessary for considering additional driving factors and for studying regions
with unique hydrological processes.
The impact of vegetation change should first be fully considered when studying the
variability of *ET*. Vegetation change significantly affects the hydrological cycle through
rainfall interception, evapotranspiration, and infiltration (Rodriguez-Iturbe, 2000;
Zhang et al., 2016b). Higher vegetation coverage increases *ET* and reduces the ratio of
$Q_r$ to *P* (Feng et al., 2016). However, most of the existing studies on *ET* variance
decomposition either ignored the effects of vegetation change or did not quantify its
contributions. Vegetation change is closely related to the Budyko controlling
parameters, and several empirical relationships have been successfully developed   on
long-term and inter-annual scales (Li et al., 2013; Liu et al., 2018; Ning et al., 2020; Xu
et al., 2013; Yang et al., 2009). However, the relationship between vegetation and its
controlling parameters on a finer time-scale has received less attention. As such, it is
important to quantitatively investigate the contribution of vegetation change to *ET*
variability on a finer time-scale.
Second, for snow-dependent regions, the short-term water balance equation was the
foundation of decomposing *ET*/or $Q_r$ variance. Its general form can be expressed as:

$$P = ET + Q_r + \Delta S, \tag{1}$$


where $P$, including liquid (rainfall) and solid (snowfall) precipitation, is the total water
source of the hydrological cycle. But this equation is unsuitable for regions where the
land-surface hydrology is highly dependent on the winter mountain snowpack and
spring snowmelt runoff. It has been reported that annual $Q_r$ originating from snowmelt
accounts for 20–70% of the total runoff, including west United States (Huning and
AghaKouchak, 2018), coastal areas of Europe (Barnett et al., 2005), west China (Li et
al., 2019b), northwest India (Maurya et al., 2018), south of the Hindu Kush (Ragettli et
al., 2015), and high-mountain Asia (Qin et al., 2020). In these regions, the mountain
snowpack serves as a natural reservoir that stores cold-season $P$ to meet the warm-
season water demand (Qin et al., 2020; Stewart, 2009). Thus, the water balance equation
should be modified to consider the impacts of snowmelt on runoff in short-term time
scale:
$$R + Q_m = ET + Q_r + \Delta S, \qquad (2)$$
where $R$ is the rainfall, and $Q_m$ is the snowmelt runoff. Many observations and
modelling experiments have found that due to global warming, increasing temperatures
would induce earlier runoff in the spring or winter and reduce the flows in summer and
autumn (Barnett et al., 2005; Godsey et al., 2014; Stewart et al., 2005; Zhang et al.,
2015). Therefore, the role of snowmelt change on $ET$ variability in snow-dependent
basins on a finer time-scale should be studied.
The overall objective of this study was to decompose the *ET* variance into the temporal
variability of multiple factors considering vegetation and snowmelt change. The six
cold alpine basins in the Qilian Mountains of northwest China were taken as an example
study area. Specifically, we aimed to: (i) determine the dominant driving factor
controlling the *ET* variance; (2) investigate the roles of vegetation and snowmelt change
in the variance; and (3) understand the interactions among the controlling factors in *ET*
variance. The proposed method will help quantify the hydrological response to changes
in snowmelt and vegetation in snowmelt-dependent regions, and our results will prove
to be insightful for water resource management in other similar regions worldwide.
**2 Materials**
**2.1 Study area**
Six sub-basins located in the upper reaches of the Heihe, Shiyang, and Shule rivers in
the Qilian Mountains were chosen as the study area (Figure 1). They are important
inland rivers in the dry region of northwest China. The runoff generated from the upper
reaches contributes to nearly 70% of the water resources of the entire basin and thus
plays an important role in supporting agriculture, industry development, and ecosystem
maintenance in the middle and downstream rivers (Cong et al., 2017; Wang et al.,
2010a). Snowmelt and in-mountain-generated rainfall make up the water supply system
for the upper basins (Matin and Bourque, 2015), and the annual average *P* exceeds 450
mm in this region. At higher altitudes, as much as 600–700 mm of $P$ can be observed
(Yang et al., 2017). Nearly 70% of the total rainfall concentrates between June and
September, while only 19% of the total rainfall occurs from March to June. Snowmelt
runoff is an important water source (Li et al., 2012; Li et al., 2018; Li et al., 2016); in
the spring, 70% of the runoff is supplied by snowmelt water (Wang and Li, 2001).
Characterised by a continental alpine semi-humid climate, alpine desert glaciers, alpine
meadows, forests, and upland meadows are the predominant vegetation distribution
patterns (Deng et al., 2013). Furthermore, this region has experienced substantial
vegetation changes and resultant hydrological changes in recent decades (Bourque and
Mir, 2012; Du et al., 2019; Ma et al., 2008).

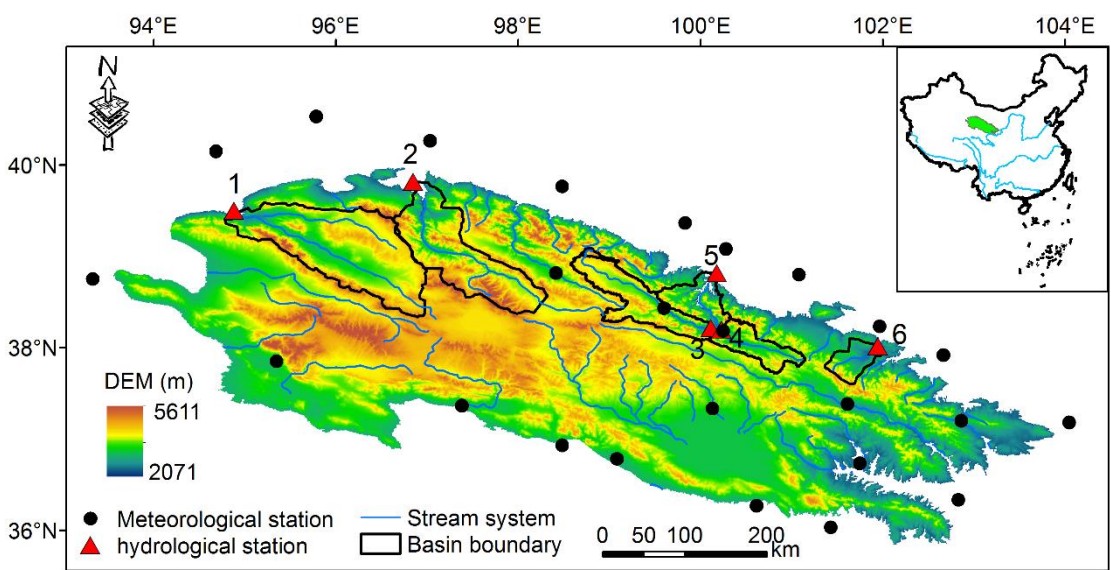


Figure 1 The six basins in China's northern Qilian Mountains. The Digital elevation data, at
30 m resolution, was provided by the Geospatial Data Cloud site, Computer Network Information

Center, Chinese Academy of Sciences.

## 2.2 Data

Daily climate data were collected for 25 stations distributed in and around the Qilian Mountains from the China Meteorological Administration. They comprised rainfall, air temperature, sunshine hours, and relative humidity and would be used to calculate the monthly $E_0$ using the Priestley and Taylor (1972) equation.

The monthly runoff at the Dangchengwan, Changmabu, Zhamashike, Qilian, Yingluoxia, and Shagousi hydrological stations were obtained for 2001–2014 from the Bureau of Hydrology and Water Resources, Gansu Province. The sum of the monthly soil moisture and plant canopy surface water with a resolution of $0.25^{\circ} \times 0.25°$ from the Global Land Data Assimilation System (GLDAS) Noah model was used to estimate the total water storage. The monthly $\Delta S$ was calculated as the water storage difference between two neighbouring months. Eight-day composites of the MODIS MOD10A2 Version 6 snow cover product from the MODIS TERRA satellite were used to produce the monthly snow cover area ($SCA$) of each basin. The $SCA$ data were used to drive the snowmelt runoff model.

A monthly normalised difference vegetation index ($NDVI$) at a spatial resolution of 1 km from the MODIS MOD13A3.006 product was used to assess the vegetation coverage ($M$), which can be calculated from the method of Yang et al. (2009):

$$M = \frac{NDVI - NDVI_{min}}{NDVI_{max} - NDVI_{min}} \tag{3}$$

where $NDVI_{max}$ and $NDVI_{min}$ are the $NDVI$ values of dense forest (0.80) and bare soil

(0.05).

$ET$ from dataset of "ground truth of land surface evapotranspiration at regional scale in
the Heihe River Basin (2012-2016) $ET_{map}$ Version 1.0" (hereafter "$ET_{map}$"), was used
to validate the reliability of our estimated $ET$. This dataset was published by National
Tibetan Plateau Data Center. It was upscaled from 36 eddy covariance flux tower sites
(65 site years) to the regional scale with five machine learning algorithms, and then
applied to estimate $ET$ for each grid cell (1 km × 1 km) across the Heihe River Basin
each day from May to September over the period 2012–2016. It has been evaluated to
have high accuracy (Xu et al., 2018). Basins 3,4,5 in our study belongs to the headwater
sub-basins of Heihe River, and our monthly $ET$ from May to September during 2012-
2014 was thus compared with $ET_{map}$.
**3 Methods**
**3.1 The Budyko framework at monthly scales**
Probing the $ET$ variability in the growing season can provide basic scientific reference
points for agricultural activities and water resource planning and management (Li et al.,
2015; Wagle and Kakani, 2014). Thus, we focus on the growing season $ET$ variability
on a monthly scale in this study.
Among the mathematical forms of the Budyko framework, this study employed the
function proposed by Choudhury (1999) and Yang et al. (2008) to assess the basin water
balance for good performance (Zhou et al., 2015):
$$ET = \frac{P_e \times E_0}{(P_e{}^n + E_0^n)^{1/n}},\tag{4}$$

where $n$ is the controlling parameter of the Choudhury–Yang equation. $P_e$ is the total
available water supply for $ET$. In previous studies, $P_e$ included $P$ and $\Delta S$ ($P_e = P - \Delta S$) on
finer time scale (Liu et al., 2019; Zeng and Cai, 2015; Zhang et al., 2016a). But
snowmelt runoff should also be considered in the snow-dependent basins. Thus, $P_e$ can
be defined as:
$$P_e = R + Q_S - \Delta S.\tag{5}$$

Equation 4 can thus be redefined as follows:
$$ET_i = \frac{(R_i + Q_{S_i} - \Delta S_i) \times E_{0_i}}{((R_i + Q_{S_i} - \Delta S_i)^{n_i} + E_{0_i}^{n_i})^{1/n_i}},\tag{6}$$

where $i$ indicates each month of the growing season (April to September). After
estimating the monthly $ET$ of the growing season using Equation 2, the values of $n$ for
each month can be obtained via Equation 6.
**3.2 Estimating the equivalent of snowmelt runoff**
With the developed relationship between snowmelt and air temperature (Hock, 2003),
the degree-day model simplifies the complex processes and performs well, so it is
widely used in snowmelt estimation (Griessinger et al., 2016; Rice et al., 2011;
Semadeni-Davies, 1997; Wang et al., 2010a). This study estimated the monthly $Q_s$ using
the degree-day model following the Wang et al. (2015) procedure. Specifically, the
water equivalent of snowmelt ($W$, mm) during the period $m$ can be calculated as:
$$\sum_{i=1}^{m} W_i = DDF \sum_{i=1}^{m} T_i^+, \tag{7}$$
where $DDF$ denotes the degree-day factor (mm/day · °C), and $T^+$ is the sum of the
positive air temperatures of each month. After obtaining $W$, the monthly $Q_s$ of each
elevation zone can be expressed as:
$$\sum_{i=1}^{m} Q_{Si} = \sum_{i=1}^{m} W_i \, SCA_i, \tag{8}$$
where $SCA_i$ is the snow cover area of each elevation zone.
According to Gao et al. (2011), the $DDF$ values of Basins 1–6 were set to 3.4, 3.4, 4.0,
4.0, 4.0, and 1.7 mm/day · °C, respectively. The six basins were divided into seven
elevation zones with elevation differences of 500 m. The sum of $Q_s$ in each elevation
zone could be considered as the total $Q_s$ of each basin. Previous studies have found that
the major snow melting period is from March to July in this area (Wang and Li, 2005;
Wu et al., 2015); furthermore, the MODIS snow product also showed that the $SCA$
decreased significantly at the end of July. Thus, the snowmelt runoff from April to July
for the growing season was estimated in this study.

## 3.3 Relationship between the Budyko controlling parameter and vegetation change

The relationships between the monthly parameters $n$ and $M$ for each basin in the growing season for 2001–2014 are presented in Figure 2. It can be seen that parameter $n$ was significantly positively related to $M$ in all six basins ($p < 0.05$), which means that $ET$ increased with increasing vegetation conditions under the given climate conditions.

In Equation 6, when $n \rightarrow 0$, $ET \rightarrow 0$, which means $M$ should have the following limiting conditions: if $ET \rightarrow 0$, $T \rightarrow 0$ (transpiration), and thus $M \rightarrow 0$. Considering the relationship shown in Figure 2 and the above limiting conditions, the general form of parameter $n$ can be expressed by power function followed previous studies (Liu et al., 2018; Ning et al., 2017; Yang et al., 2007):

$$n = a \times M^b, \tag{9}$$

where $a$ and $b$ are constants, and their specific values for each basin are fitted in Figure 2.

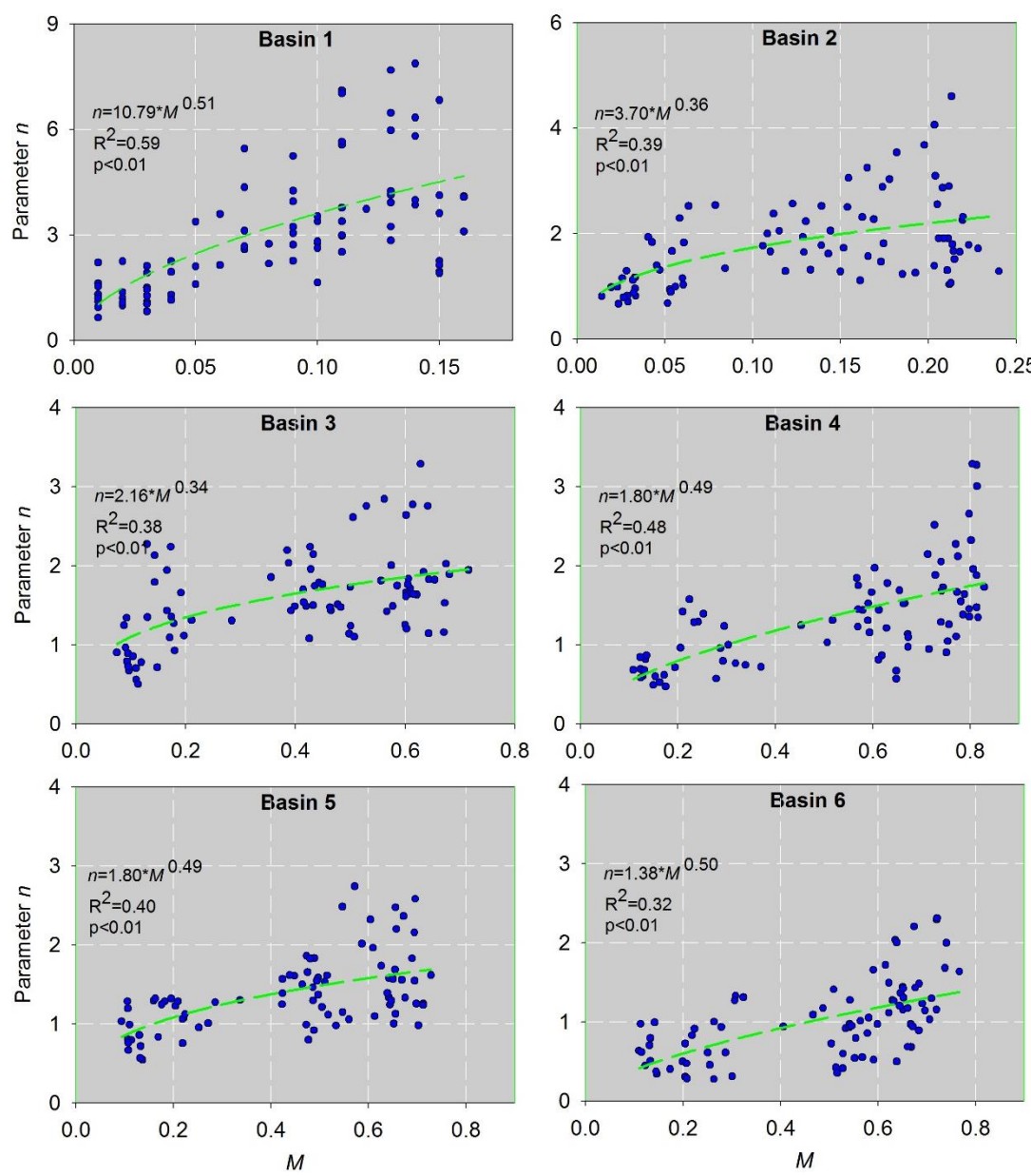

Figure 2 Relationships between the parameter $n$ and the vegetation coverage for each basin on a monthly scale.

### 3.4 *ET* variance decomposition

Liu et al. (2019) proposed a framework to identify the driving factors behind the temporal variance of $Q_r$ by combining the unbiased sample variance of $Q_r$ with the total

differentiation of $Q_r$ changes. Here, we extended this method by considering the effects
of changes in snowmelt runoff and vegetation coverage on $ET$ variance.
By combining Equation 6 with Equation 9, Equation 6 can be simplified as $ET \approx f(R_i,$
$Q_{mi}, \Delta S_i, E_{0i}, M_i)$. Thus, the total differentiation of $ET$ changes can be expressed as:

$$dET_i = \frac{\partial f}{\partial R}dR_i + \frac{\partial f}{\partial Q_s}dQ_{m_i} + \frac{\partial f}{\partial \Delta S}d\Delta S_i + \frac{\partial f}{\partial E_0}dE_{0_i} + \frac{\partial f}{\partial M}dM_i + \tau, \qquad (10)$$

where $\tau$ is the error. $\frac{\partial f}{\partial R}, \frac{\partial f}{\partial Q_m}, \frac{\partial f}{\partial \Delta S}, \frac{\partial f}{\partial E_0}, \frac{\partial f}{\partial M}$ are the partial differential coefficients of
$ET$ to $R, Q_m, \Delta S, E_0$ and $M$ , respectively, which can be calculated as:

$$\frac{\partial ET}{\partial R} = \frac{\partial ET}{\partial Q_m} = -\frac{\partial ET}{\partial \Delta S} = \frac{ET}{P_e} \times \left(\frac{E_0^n}{P_e^n + E_0^n}\right), \qquad (11a)$$

$$\frac{\partial ET}{\partial E_0} = \frac{ET}{E_0} \times \left(\frac{P_e^n}{P_e^n + E_0^n}\right), \qquad (11b)$$

$$\frac{\partial ET}{\partial M} = \frac{ET}{n}\left(\frac{\ln(P_e^n + E_0^n)}{n} - \frac{P_e^n \ln P + E_0^n \ln E_0}{P_e^n + E_0^n}\right) \times a \times b \times M^{b-1}. \qquad (11c)$$

The first-order approximation of $ET$ changes in Equation 10 can be expressed as:

$$\Delta ET_i \approx \varepsilon_1 \Delta R_i + \varepsilon_2 \Delta Q_{s_i} + \varepsilon_3 \Delta S_i + \varepsilon_4 \Delta E_{0_i} + \varepsilon_5 \Delta M_i, \qquad (12)$$

where $\varepsilon_1 = \frac{\partial ET}{\partial R}$; $\varepsilon_2 = \frac{\partial ET}{\partial Q_s}$; $\varepsilon_3 = \frac{\partial ET}{\partial \Delta S}$; $\varepsilon_4 = \frac{\partial ET}{\partial E_0}$; $\varepsilon_5 = \frac{\partial ET}{\partial M}$.
In this study, the temporal variance of $ET$ reflects the fluctuation of monthly $ET$ in
growing season for years, which can be quantified by the unbiased sample variance
$(\sigma_{ET}^2)$ :

$$\sigma_{ET}^2 = \frac{1}{N-1}\sum_{i=1}^{N}(ET_i - \overline{ET})^2 = \frac{1}{N-1}\sum_{i=1}^{N}(\Delta ET_i)^2. \qquad (13)$$

where $\overline{ET}$ is the long term monthly mean of $ET$. $N$ is the sample size, it equals 84 in
this study (6 months/year×14 years=84 months). $i$ is used to index time series of month
from 1 to $N$.   $\sigma_{ET}^2$ indicates how far a set of monthly $ET$ in growing season is spread
out from their average value. The larger $\sigma_{ET}^2$, the larger fluctuation of $ET$, and vice
versa.
Combining Equation 12 with Equation 13, $\sigma_{ET}^2$ can be decomposed as the contribution
from different variance/covariance sources:

$$\sigma_{ET}^2 = \sum_{i=1}^{N}(\varepsilon_1\Delta R_i + \varepsilon_2\Delta Q_{s_i} + \varepsilon_3\Delta S_i + \varepsilon_4\Delta E_{0_i} + \varepsilon_5\Delta M_i)^2. \qquad (14)$$

Expanding Equation 14, $\sigma_{ET}^2$ can be further rewritten as:
$\sigma_{ET}^2 = \varepsilon_1^2\sigma_R^2 + \varepsilon_2^2\sigma_{Q_s}^2 + \varepsilon_3^2\sigma_{\Delta S}^2 + \varepsilon_4^2\sigma_{E_0}^2 + \varepsilon_5^2\sigma_M^2 + 2\varepsilon_1\varepsilon_2\text{cov}(R,Q_s) +$
$2\varepsilon_1\varepsilon_3\text{cov}(R,\Delta S) + 2\varepsilon_1\varepsilon_4\text{cov}(R,E_0) + 2\varepsilon_1\varepsilon_5\text{cov}(R,M) + 2\varepsilon_2\varepsilon_3\text{cov}(Q_s,\Delta S) +$
$2\varepsilon_2\varepsilon_4\text{cov}(Q_s,E_0) + 2\varepsilon_2\varepsilon_5\text{cov}(Q_s,M) + 2\varepsilon_3\varepsilon_4\text{cov}(E_0,\Delta S) + 2\varepsilon_3\varepsilon_5\text{cov}(M,\Delta S) +$
$2\varepsilon_4\varepsilon_5\text{cov}(E_0,M),$               (15)
where $\sigma$ represents the standard deviation, and $cov$ represents the covariance. Equation
15 can be further simplified as:
$\sigma_{ET}^2 = F(R) + F(Q_s) + F(\Delta S) + F(E_0) + F(M) + F(R\_Q_s) + F(R\_\Delta S) +$
$F(R\_E_0) + F(R\_M) + F(Q_s\_\Delta S) + F(Q_s\_E_0) + F(Q_s\_M) + F(\Delta S\_E_0) +$
$F(\Delta S\_M) + F(E_0\_M),$                 (16)
Where $F$ is the individual contributions of each factor; each two factors linked by
underscore represents the interaction effects between them.
By separating out Equation 16, the contribution of each factor to $\sigma_{ET}^2$ can be calculated
as:
$$C(X_\mathrm{j}) = \frac{F(X_j)}{\sigma_{ET}^2} \times 100\%, \tag{17}$$
where $C(X_\mathrm{j})$ is the contribution of factor $F(j)$ to $\sigma_{ET}^2$, and $j = 1$–15, representing the 15
factors in Equation 16.
**4 Results and Discussion**
**4.1 The effects of monthly storage change and snowmelt runoff in the Budyko**
**framework**
The Budyko framework is usually used for analyses of long-term average catchment
water balance; however, it was employed for the interpretation of the monthly
variability of the water balance in this study. Thus, it's very necessary to validate the
feasibility of Budyko equation for monthly variability. Furthermore, the impact of $\Delta S$
on the representation of Budyko framework on a finer time-scale has been assessed
by several studies (Chen et al., 2013; Du et al., 2016; Liu et al., 2019; Zeng and Cai,
2015). However, the impact of $Q_m$ and its combined effects with $\Delta S$ in snowmelt-
dependent basins are mostly ignored. Therefore, we present the water balance in the
monthly scale of six basins in the Budyko's framework with three different
computations of aridity index ($\phi = E_0/P_e$) or $ET$ ratio ($ET/P_e$) in Figure 3. In Figure 3a,
$ET = R - Q_r$ when $R$ is considered as water supply, i.e., $P_e = R$. The points of monthly $ET$
ratio and aridity index in April and May were well below Budyko curves in 6 basins;
monthly $ET$ ratio was even negative in several year, which means the local rain are not
the only sources of $ET$ in this area, especially in spring. In Figure 3b, $ET = R - \Delta S - Q_r$ with
$P_e = R - \Delta S$. Compared with figure 3a, the way-off points in April and May were improved
to a certain extent but negative points still existed, suggesting that except for $R$, $\Delta S$ also
play a significant role in maintaining spring $ET$, but the variability of $ET$ cannot be
completely explained by these two variables. In Figure 3c, $ET = R - \Delta S + Q_m - Q_r$ with
$Pe = R - \Delta S + Q_m$. Compared to the points in Figures 3a-b, all points focused on Budyko's
curves more closely in each basin when $Pe = R + Q_m - \Delta S$. From this comparison, it can be
concluded that the Budyko framework is applicable to the monthly scale in snowmelt-
dependent basins, if the water supply is described accurately by considering $\Delta S$ and $Q_m$.

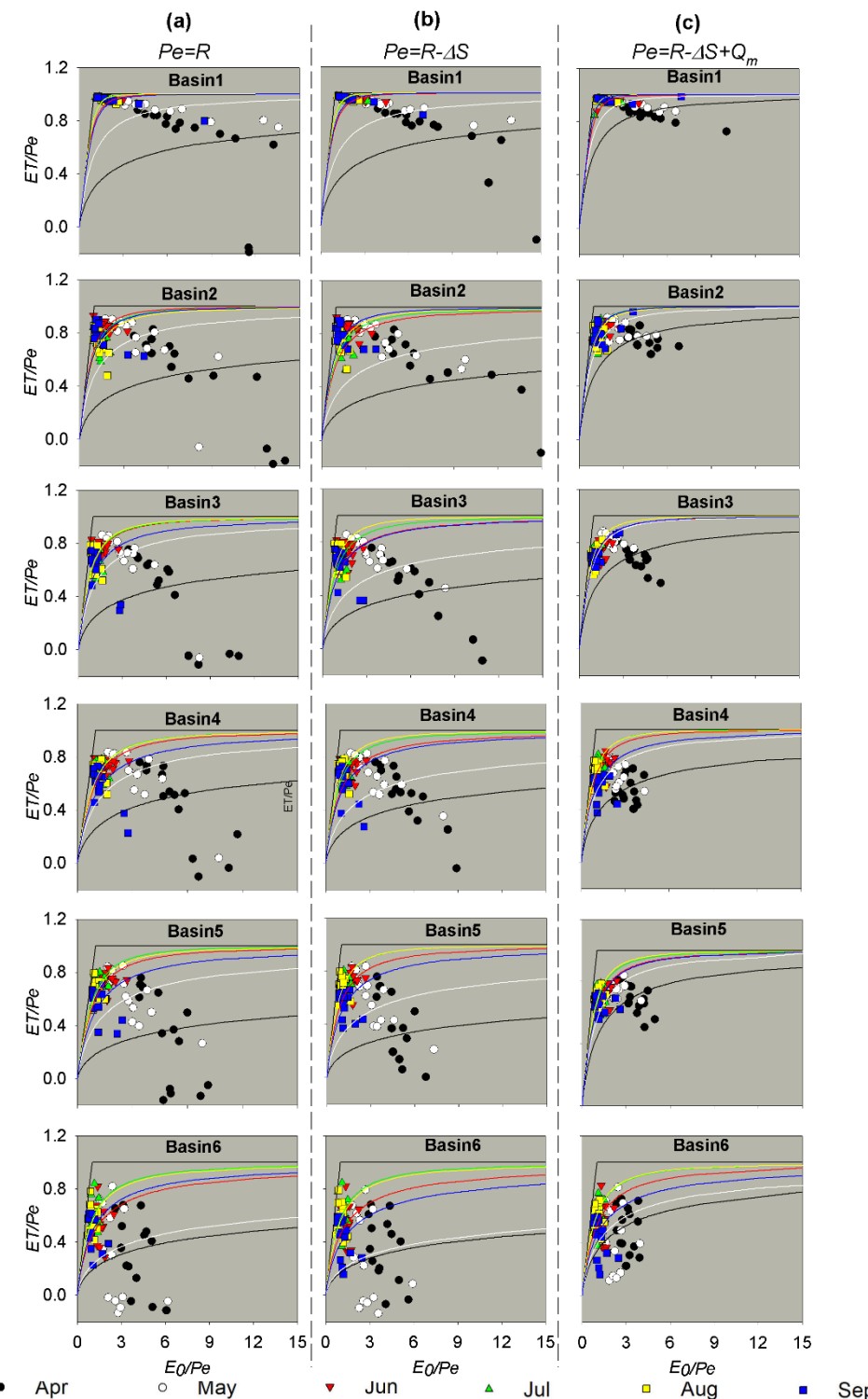


Figure 3 Plots for the aridity index vs. evapotranspiration index scaled by the available water

supply for monthly series in the growing season. The total water availability is (a) $R$, (b) $R - \Delta S$,

(c) $R + Q_m - \Delta S$. The $n$ value for each Budyko curve is fitted by long-term averaged monthly data.

## 4.2 Variations in the growing season water balance

The mean and standard deviation ($\sigma$) for each item in the growing season water balance in the six basins are summarised in Tables 1 and 2. The proportion of $\Delta S$ in the water balance was small, with a mean value of 1.2 mm; however, its intra-annual fluctuation was relatively large, with a $\sigma_{\Delta S}$ of 5.3 mm, and $\sigma_{\Delta S}$ was even as high as 9.0 mm in Basin 6. Compared to $\Delta S$, $Q_m$ represented a larger proportion of the water balance with a mean of 8.5±6.5 mm, indicating its important role in the basin water supply. For this region, the water supply of $ET$ was not only $R$ but also included $Q_m$ and $\Delta S$. Consequently, the mean monthly $ET$ generally approached $R$ (55.8±27.4 mm) or higher values in Basin 1.

Table 1 Averaged monthly hydrometeorological characteristics and vegetation coverage in the growing season (2001–2014).

| ID | Station | Area | $R$ | $Q_m$ | $\Delta S$ | $E_0$ | $M$ | $n$ | $E$ |
|----|---------|------|-----|-------|-----------|-------|-----|-----|-----|
| 1 | Dangchengwan | 14325 | 57.2 | 8.6 | 0.7 | 126.7 | 0.08 | 3.08 | 59.1 |
| 2 | Changmabu | 10961 | 68.9 | 10.8 | 1.1 | 123.0 | 0.13 | 1.79 | 59.3 |
| 3 | Zhamashike | 4986 | 73.5 | 10.6 | 1.5 | 120.3 | 0.40 | 1.59 | 59.1 |
| 4 | Qilian | 2452 | 74.5 | 9.0 | 1.4 | 116.8 | 0.44 | 1.37 | 54.9 |
| 5 | Yingluoxia | 10009 | 77.2 | 7.4 | 1.1 | 117.4 | 0.53 | 1.35 | 55.1 |
| 6 | Shagousi | 1600 | 83.5 | 4.8 | 1.4 | 116.3 | 0.48 | 1.01 | 47.1 |

The change patterns of the monthly $R$, $\Delta S$, $Q_m$, and $ET$ during the growing season are presented in Figure 4 and Supplementary Figures S1–S3. $R$ exhibited a regular unimodal trend, with a maximum value occurring in July. The maximum $Q_m$ appeared

in May, which is a result that is in agreement with previous studies in this region (Wang
and Qin, 2017; Zhang et al., 2016c). The peak of $\Delta S$ lagged that of $Q_m$ for one month
in Basins 1–4 and three months in Basins 5–6, indicating a recharge of soil water by
snowmelt. Yang et al. (2015) also detected the time differences between $\Delta S$ and $Q_m$ and
found that $\Delta S$ had a time lag of 3–4 months more than did $Q_m$ in the Tarim River Basin,
another arid alpine basin in north-western China with hydroclimatic conditions similar
to those of the study region. Further, the abundant $R$ in July should contribute to more
available water for $\Delta S$; however, the $\Delta S$ in July was relatively small. This can be
partially explained by the higher water consumption, i.e. the $ET$ in July. In a manner
similar to the change pattern of $R$, $ET$ exhibited a unimodal trend, suggesting the crucial
role of $R$.

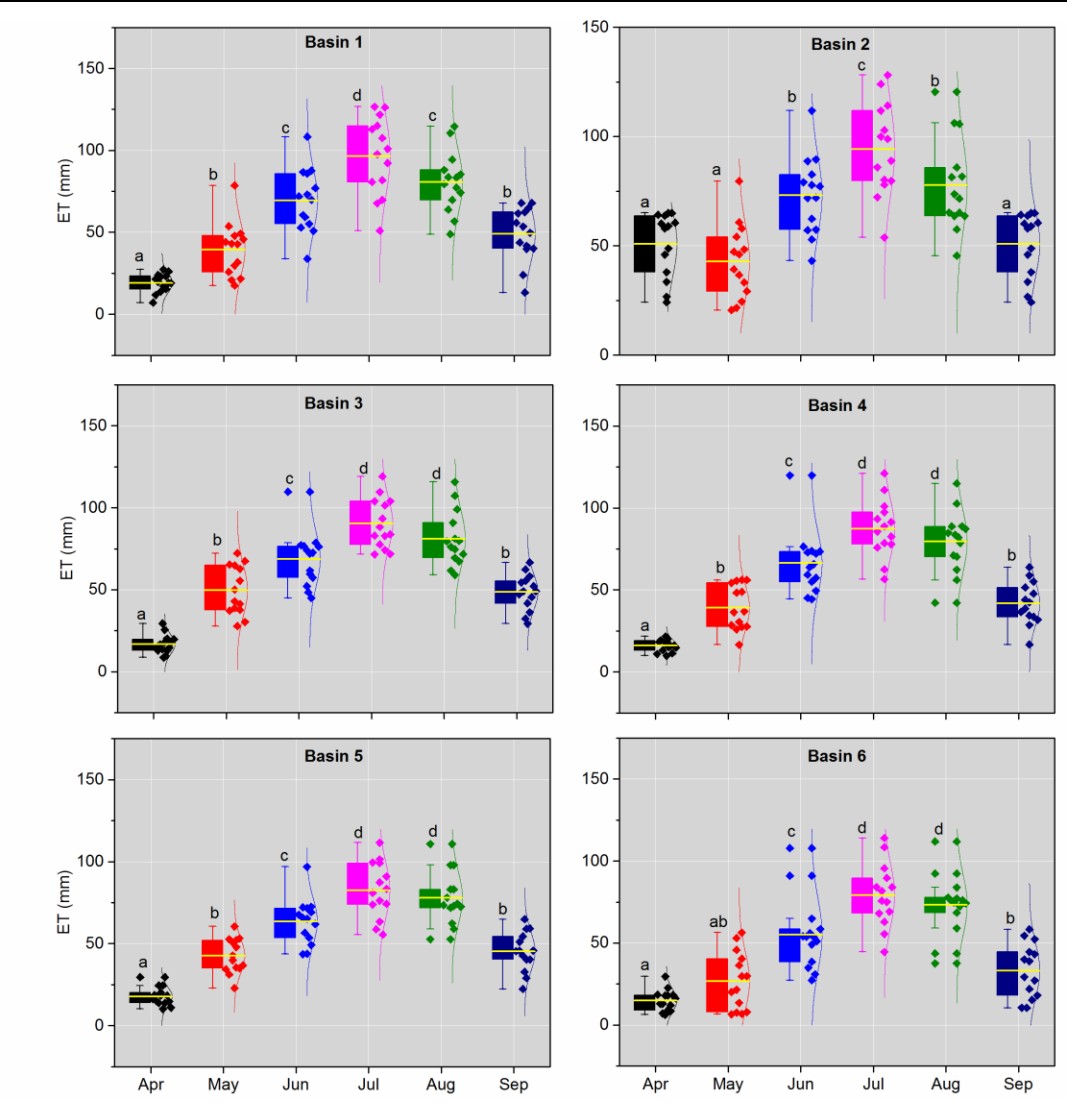

Figure 4 Variations in the monthly *ET* for each basin during 2001–2014. A distribution curve is

shown to the right side of each box plot, and the data points are represented by diamonds.

Different letters indicate significant differences at p < 0.05.

## 4.3 Controlling factors of the *ET* variance

The contributions of *R*, $E_0$, $Q_m$, $\Delta S$, and *M* to $\sigma_{ET}^2$ for each basin are shown in Figure

5. The results showed that the variance of these five factors could explain $\sigma_{ET}^2$, with the

total contribution rates ranging from 56.5% (Basin 6) to 98.6% (Basin 1). With the

decreasing $\phi$ from Basin 1 to Basin 6, C($R$) showed an increasing trend, ranging from
40.6% to 94.2%; conversely, C($E_0$) exhibited a decreasing trend, ranging from 0.2% to
4.1%. This result indicated that $R$ played a key role in $\sigma_{ET}^2$ in this region. Similarly,
Zhang et al. (2016a) found that C($P$) increased rapidly with increasing $\phi$, whereas C($E_0$)
decreased rapidly based on 282 basins in China. Our results are also consistent with
previous conclusions that changes in $ET$ or $Q_r$ are dominated by changes in water
conditions rather than by energy conditions in dry regions (Berghuijs et al., 2017; Yang
et al., 2006; Zeng and Cai, 2016; Zhang et al., 2016a).
The $M$ variance had the second largest contribution to $\sigma_{ET}^2$ with a mean C($M$) value of
4.3% for the six basins. Specifically, C($M$) showed an increasing trend from 0.5% to
9.5% with the decreasing $\phi$, implying that the contribution of vegetation change to $ET$
variance was larger in relatively humid basin. It can be explained that transpiration is
more sensitive to vegetation change, and thus the higher vegetation coverage could
increase the proportion of transpiration to $ET$ in humid regions (Niu et al., 2019; Zhang
et al., 2020). The Budyko hypothesis stated that change in $ET$ is controlled by change
in available energy when water supply is not a limiting factor under humid conditions
(Budyko, 1974; Yang et al., 2006). The increasing $M$ results in the reallocation of
available energy between canopy and soil. Specifically, more energy is consumed by
canopy thus increases transpiration. Further, Previous studies have found that $ET$ differs
greatly among species, because of the difference in canopy roughness, the timing of
physiological functioning, water holding capacity of the soil and rooting depth of the
vegetation (Baldocchi et al., 2004; Bruemmer et al., 2012). Generally, forest had larger
$ET$ than grassland (Ma et al., 2020; Zha et al., 2010). The fraction of forest area is
relatively high and thus lead to the higher contributions to $ET$ for whole basin in the
humid region. For example, Wei et al. (2018) showed that the global average variation
in the annual $Q_r$ due to the vegetation cover change was 30.7±22.5% in forest-
dominated regions on long-term scales, which was higher than our results because of
their higher forest cover.
The contribution of the $Q_m$ variance ranked third with a mean value of 1.8%. Similar as
C($R$), C($Q_m$) showed a downward trend with the decreasing $\phi$, ranging from 2.9% to
0.4%. The larger C($Q_m$) can be explained by the larger variance in $Q_m$ in Basins 2–4 (σ
values in Table 2). However, the $Q_m$ in Basin 1 was only 8.6 mm, and C($Q_m$) was the
largest in all six sub-basins (2.9%). It can be explained that the contribution of each
variable to $\sigma_{ET}^2$ was not only the product of the partial differential coefficients, but also
relied on its variance value according to Equation 14. Specifically, the partial
differential coefficients of 0.1 for a variable means that a 10% change in that variable
may result in a change in $ET$ by 1%, which can only reflect the theoretical contribution
of each variable. By multiplying the variance value, the actual contribution of each
variable could be obtained. The $\varepsilon_{Q_m}$ value was the largest in Basin 1 and thus led to the
largest C($Q_m$). In addition, shifts in the snowmelt period can also partially explain the
positive contribution of the $Q_m$ variance. Like many snow-dominated regions of the
world (Barnett et al., 2005), climate warming shifted the timing of snowmelt earlier in
the spring in the Qilian Mountains (Li et al., 2012). Earlier snowmelt due to a warmer
atmosphere resulted in increased soil moisture and a greater proportion of $Q_m$ to $ET$
(Barnhart et al., 2016; Bosson et al., 2012).
Previous studies have considered that most precipitation changes are transferred to
water storage (Wang and Hejazi, 2011); thus, $\Delta S$ has distinct impacts on the intra-annual
$ET$ or $Q_r$ variance in arid regions (Ye et al., 2015; Zeng and Cai, 2016; Zhang et al.,
2016a). However, the study region under investigation has a small $C(\Delta S)$ with a mean
value of 1.02%, which is likely to be caused by the vegetation conditions and time-
scale. First, the six basins have higher vegetation coverage compared to other arid
basins; consequently, plant transpiration and rainfall interception consume most of the
water supply and reduce the transformation of rainfall to water storage. This is
consistent with previous studies that showed that the fractional contribution of
transpiration to $ET$ would increase with increasing woody cover (Villegas et al., 2010;
Wang et al., 2010b). Second, the large contribution of $\Delta S$ to the intra-annual $ET$ or $Q_r$
variance in arid regions is mostly detected at monthly scales. The smaller $\Delta S$ in the non-
growing season will increase the annual value of $\sigma_{\Delta S}$. However, this study focused on
the growing season with a smaller $\sigma_{\Delta S}$, which consequently led to a lower $C(\Delta S)$.

**4.4 Interaction effects between controlling factors on the *ET* variance**

The interaction effect of two factors on the *ET* variance was represented by their covariance coefficients using Equations 15 and 16 (Figure 5). Among the ten groups of interaction effects, the coupled *R* and *M* had the largest contribution to the *ET* variance, with a mean value of 24.3%. The positive covariance of *R* and *M* indicated that *M* changes in-phase with *R* (i.e. *R* occurred in the growing season), thus increasing the *ET* variance. C(*R_M*) showed an increasing trend from 9.9% to 34.6% with decreasing $\phi$. With different water conditions, the types and proportions of the main ecosystems varied across basins. In particular, *F* showed an increasing trend with decreasing $\phi$, which partially explained the spatial variations in C(*R_M*). Previous studies concluded that the differences in physiological and phenological characteristics of ecosystem types are likely to modulate the response of the ecosystem *ET* to climate variability (Bruemmer et al., 2012; Falge et al., 2002; Li et al., 2019a). For example, Yuan et al. (2010) found that, at the beginning of the growing season, a significantly higher *ET* was observed in evergreen needleleaf forests; however, during the middle term of the growing season (June–August), the *ET* was largest in deciduous broadleaf forests in a typical Alaskan basin.

As an indicator of climate seasonality, the covariance of *R* and $E_0$ indicates matching conditions between the water and energy supplies, such as the phase difference between the storm season and warm season. A positive cov($R$, $E_0$) suggests an in-phase R change

with $E_0$ and consequently increases the *ET* variance. In this study, following C(*R_M*),
the coupled *R* and $E_0$ had a large impact on the *ET* variance with a mean contribution
of 14.1%. With a typical temperate continental climate, the study area has in-phase
water and energy conditions; however, its *ET* is limited by the water supply in spite of
the abundant energy supply (Yang et al., 2006). The vegetation receives the largest
water supply in the growing season and can vary its biomass seasonally in order to
adapt to the *R* seasonality (Potter et al., 2005; Ye et al., 2016). Consequently, the impact
of climate variability on *ET* variance was mainly reflected by the *R* seasonality in the
study area.
In comparison, the interacting effects between *R* and $Q_m$, *M* and $Q_m$, *R* and $\Delta S$, and $Q_m$
and $E_0$ contributed negatively to the *ET* variance. Among them, the effect of the coupled
*R* and $Q_m$ was largest with a C(*R_$Q_m$*) of −7.6%. This may suggest that $Q_m$ changes
were out-of-phase with *R*. Specifically, the major snow melting period was from March
to May, when snowmelt water accounts for ~70% of the water supply; however, ~ 65%
of the annual *R* occurred in the summer (June–August) (Li et al., 2019a). Overall, $Q_m$
sustains the *ET* in the spring, but *R* supports the *ET* in the summer.

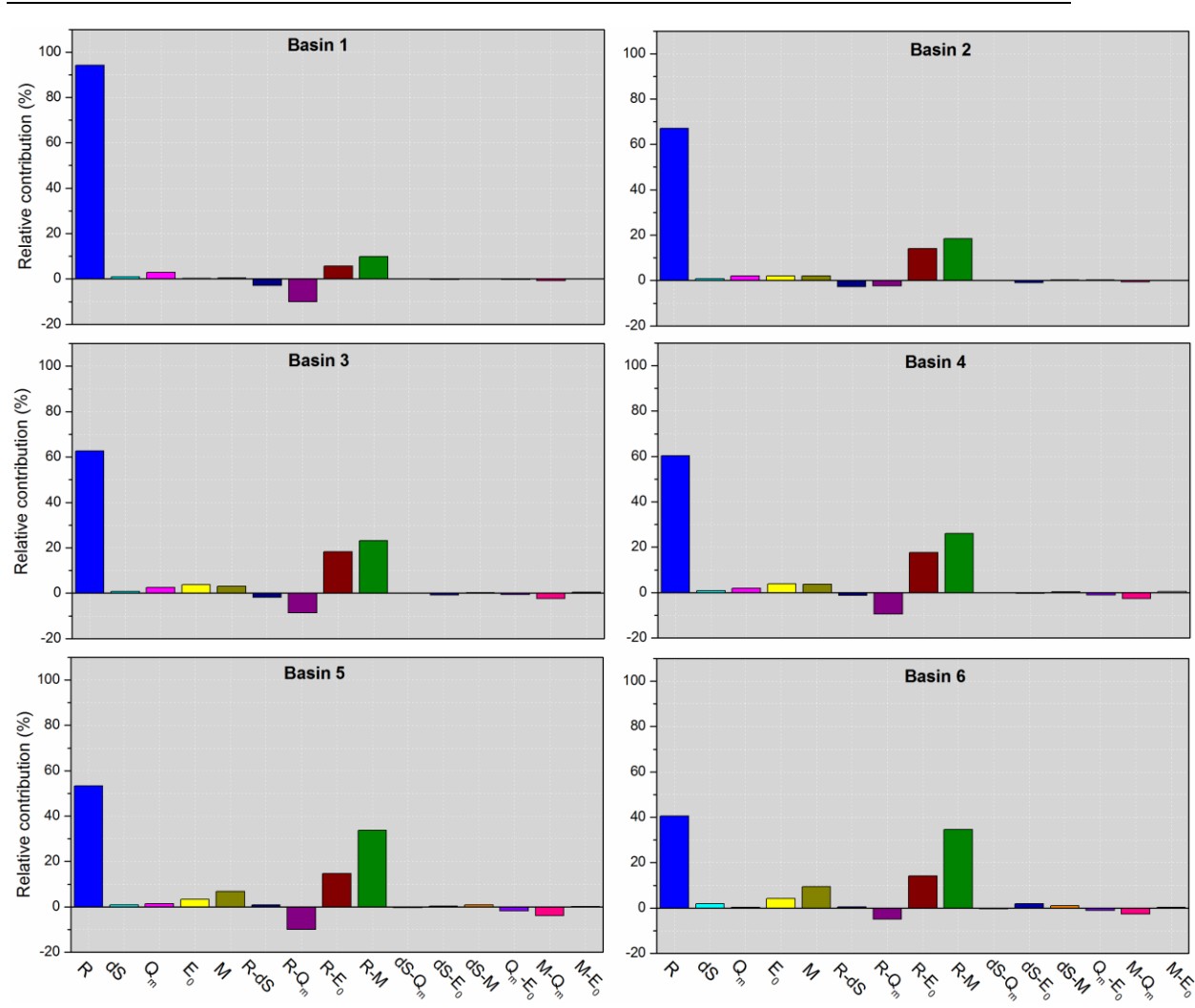


Figure 5 Contribution to the *ET* variance in the growing season from each component in Equation

15.

**4.5 Uncertainties**

Uncertainties from different sources may result in errors for this study. First, this study

estimated $\Delta S$ and $Q_m$ with the GLDAS Noah land surface model and the degree-day

model, respectively. Although the GLDAS_$\Delta S$ has been widely used in hydrological

studies, it ignores the change in deep groundwater (Nie et al., 2016; Syed et al., 2008;

Zhang et al., 2016), which may lead to errors in *ET* estimation based on water balance

equation. But previous studies showed that the groundwater change in our study area is

relatively small, and can thus be ignored. For example, Du et al. (2016) used the abcd

model to quantitatively determine monthly variations of water balance for the sub-

basins of Heihe River (including basins 3-5 in our study ) and found that the soil water

storage change have obvious effects on the monthly water balance, whilst the impact of

monthly groundwater storage change is negligible. Furthermore, it has been found that

any change in climate conditions and underlying basin characteristics will affect the

contributions of heat balance components and cause temporal variations of *DDF*

(Kuusisto, 1980; Ohmura, 2001). But previous studies indicated that there is no

significant seasonal change in *DDF* in west China (Zhang et al., 2006); as such, it is

acceptable to estimate snowmelt runoff using fixed *DDF* values in this study. In

comparison, the contribution of snow meltwater to runoff ($F_s$) was 12.9% in Basin 2

during 1971-2015 by using Spatial Processes in Hydrology model(Li et al., 2019), while

$F_s$ was 25% in Basin 3 from 2001 to 2012 based on geomorphology-based

ecohydrological model (Li et al., 2018), <10% in Basin 6 during 1961-2006 by using

SRM model (Gao et al., 2011). Our results indicated that the $F_s$ in Basin 2, 3 and 6 were

14.8%, 24.5% and 6.7%, respectively, which were close to those from different models.

Finally, the uncertainties of $\Delta S$ and $Q_m$ may lead to errors in *ET* estimation by water

balance equation. To validate the reliability of our estimated *ET*, the comparison with

$ET_{map}$ from May to September during 2012-2014 was conducted (Figure S4). The

results showed that our estimated $ET$ fitted well with $ET_{map}$ and basically fell around
the 1:1 line, indicating $ET$ estimated using water balance equation by considering the
items of $\Delta S$ and $Q_m$ is acceptable. However, it cannot be ignored that our estimated $ET$
was generally lower than $ET_{map}$. The error of rainfall spatial interpolation may explain
the underestimation of $ET$. Most meteorological stations are located at low elevations
or in river valleys, but some stations are distributed in high elevations in Qilian
Mountain (Figure 1). It has been found that rainfall in mountainous regions is generally
larger than that in plain regions (Qiang et al., 2015). Even the topography effect was
considered for interpolation, it still resulted in bias in areal rainfall. The best method to
improve the quality of spatial rainfall estimation is to increase the density of the
monitoring network. However, this process is limited by harsh environment and funds
(Buytaert et al., 2006). The errror of rainfall will be transferred to contribution
quantification of $ET$ variance by underestimating rainfall contribution, while
overestimating $Q_m$ and $\Delta S$ contribution.
Second, previous studies concluded that three main factors could be responsible for the
variability of $n$, including underlying physical conditions (such as soil and topography
characteristics) (Milly, 1994; Yang et al., 2009), climate seasonality (such as the
temporal variability of rainfall, mismatch between water and energy) (Ning et al., 2017;
Potter et al., 2005) and vegetation dynamics (Donohue et al., 2007; Zhang et al., 2001).
On the short time scale, the changes in soil and topography are negligible and its impact
on the variability of n can be ignored. In consequence, the factors, should be considered,
are climate seasonality and vegetation dynamics. When parameterizing $n$, this study
considered $M$ but ignored climate seasonality since the covariance item between $R$ and
$E_0$, i.e. $\varepsilon_1\varepsilon_4\mathrm{cov}(R, E_0)$ in the Equation (15) can represent climate seasonality. In addition,
human influence represented by parameter $n$ on the water balance cannot be ignored,
which remains further investigation.

## 5 Conclusion

Recently, several studies have applied a variance decomposition framework based on
the Budyko equation to elucidate the dominant driving factors of the $ET$ variance at
annual and intra-annual scales by decomposing the intra-annual $ET$ variance into the
variance/covariance of $P$, $E_0$, and $\Delta S$. Vegetation changes can greatly affect the $ET$
variability, but their effects on the $ET$ variance on finer time-scales was not quantified
by this decomposed method. Further, in snow-dependent regions, snowpack stores
precipitation in winter and releases water in spring; thus, $Q_m$ plays an important role in
the hydrological cycle. Therefore, it is also necessary to consider the role of the $Q_m$
changes on the $ET$ variability.
In this study, six arid alpine basins in the Qilian Mountains of northwest China were
chosen as examples. The monthly $Q_m$ during 2001–2014 was estimated using the
degree-day model, and the growing season $ET$ was calculated using the water balance
equation ($ET = R + Q_s - Q_r - \Delta S$). The controlling parameter $n$ of the Choudhury–
Yang equation was found to be closely correlated with $M$, as estimated by *NDVI* data.
Thus, by combining the Choudhury–Yang equation with the semi-empirical formula
between $n$ and $M$, the growing season $\sigma_{ET}^2$ is decomposed into the temporal variance
and covariance of $R$, $E_0$, $\Delta S$, $Q_m$, and $M$. The main results showed that considering $Q_m$
and $\Delta S$ in the water balance equation can improve the performance of the Budyko
framework in snow-dependent basins on a monthly scale; $\sigma_{ET}^2$ was primarily enhanced
by the $R$ variance, followed by the coupled $R$ and $M$ and then the coupled $R$ and $E_0$. The
enhancing effects of the variance in $M$ and $Q_m$ cannot be ignored; however, the
interactions between $R$ and $Q_m$, $M$ and $Q_m$, $R$ and $\Delta S$, and $Q_m$ and $E_0$ dampened $\sigma_{ET}^2$.
As a simple and effective method, our extended *ET* variance decomposition method has
the potential to be widely used to assess the hydrological responses to changes in the
climate and vegetation in snow-dependent regions at finer time-scales.
Table 2 The elasticity coefficients of ET for five variables and the standard deviation of each variable
for the six basins.

| | Elasticity coefficients | | | | | Standard deviation | | | | | | |
|---|---|---|---|---|---|---|---|---|---|---|---|---|
| Basin | $\varepsilon_R$ | $\varepsilon_{Q_m}$ | $\varepsilon_{\Delta S}$ | $\varepsilon_{E_0}$ | $\varepsilon_M$ | $\sigma_R$, mm | $\sigma_{Q_m}$, mm | $\sigma_{\Delta S}$, mm | $\sigma_{E_0}$, mm | $\sigma_M$ | Predicted $\sigma_{ET}$, mm | Assessed $\sigma_{ET}$, mm |
| 1 | 0.85 | 0.85 | −0.85 | 0.06 | 41.94 | 34.4 | 6.0 | 3.4 | 25.5 | 0.05 | 30.2 | 31.2 |
| 2 | 0.56 | 0.56 | −0.56 | 0.16 | 55.84 | 40.6 | 7.0 | 4.3 | 24.7 | 0.07 | 27.8 | 30.3 |
| 3 | 0.46 | 0.46 | −0.46 | 0.20 | 20.81 | 42.5 | 8.5 | 4.9 | 23.6 | 0.21 | 24.9 | 27.9 |
| 4 | 0.44 | 0.44 | −0.44 | 0.19 | 20.58 | 40.1 | 7.2 | 4.8 | 23.1 | 0.21 | 22.5 | 25.8 |
| 5 | 0.43 | 0.43 | −0.43 | 0.19 | 24.60 | 39.8 | 6.3 | 5.1 | 22.0 | 0.25 | 23.3 | 25.0 |
| 6 | 0.33 | 0.33 | −0.33 | 0.18 | 31.51 | 41.2 | 4.0 | 9.0 | 23.6 | 0.21 | 21.3 | 24.3 |



## Data availability

The Digital elevation data are available at
http://www.gscloud.cn/sources/accessdata/310?pid=302. Meteorological data are
available at
http://data.cma.cn/data/detail/dataCode/SURF_CLI_CHN_MUL_DAY_CES_V3.0.ht
ml. The runoff records were obtained from the Bureau of Hydrology and Water
Resources, Gansu Province. The GLDAS data are available at
https://disc.gsfc.nasa.gov/datasets/GLDAS_NOAH025_M_2.0/summary. MODIS
MOD10A2 Version 6 snow cover products are available at
https://nsidc.org/data/mod10a2. MODIS MOD13A3.006 products are available at
https://lpdaac.usgs.gov/products/mod13a3v006/. The dataset of "ground truth of land
surface evapotranspiration at regional scale in the Heihe River Basin (2012-2016)
ETmap Version 1.0" are available at http://data.tpdc.ac.cn/zh-hans/data/8efbb18d-
bc02-4bf6-9f21-345480d6637f/?q=ETMap.

## Author contributions

Tingting Ning: Methodology, Writing–original draft, Software, Visualisation
Zhi Li: Writing–review & editing
Qi Feng: Conceptualisation, Supervision
Zongxing Li and Yanyan Qin: Data curation, Resources
**Competing interests**
The authors declare that they have no conflicts of interest.
**Acknowledgements**
This study was supported by the National Natural Science Foundation of China
(41807160), Opening Research Foundation of Key Laboratory of Land Surface Process
and Climate Change in Cold and Arid Regions, Chinese Academy of Sciences (LPCC
2020003), the "Western Light"-Key Laboratory Cooperative Research Cross-Team
Project of Chinese Academy of Sciences, the CAS 'Light of West China' Program
(Y929651001), the Major Program of the Natural Science Foundation of Gansu
Province, China (18JR4RA002) , and the Second Tibetan Plateau Scientific Expedition
and Research Program (STEP, Grant No.2019QZKK0405).

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

1790.