# Peer review of "Attribution of growing season evapotranspiration variability considering snowmelt and vegetation changes in the arid alpine basins"

_Hydrology and Earth System Sciences, 2020_

## Referee Comment (RC1) · Anonymous Referee #1 · 8 Jan 2021

The authors extend existing Budyko-type approaches for decomposing monthly ET variance (eg Liu et al 2019) amongst variances (average monthly deviation from an annual mean value) in underlying physical drivers of plant water use (e.g. rainfall). In particular, the model extension now accounts for variance in snowmelt fluxes and variance in vegetation cover. The manuscript is a logical extension of work previously published on the topic. However, I do have serious concerns with clarity of presentation in some parts of the manuscript, as well as the underlying "consistency" of the datasets used in the study (detailed in specific comments below).

I also have a general question about the overall approach (that will probably reveal

[Figure]

my own ignorance about these methods!). When I think about "variability" in ET, I first think about the year-by-year variation in the magnitude of ET in a particular month. However, if I'm understanding this manuscript correctly, the variability that is under consideration is the average of the monthly deviation of ET (or underlying drivers) from a long-term annual average. Is this interpretation correct? That is, in Equation 12 does \overline{ET} equal the long term annual mean, and does the index "i" in this case index all of the values for a given month? If so, this is somewhat confusing, as in the previous sections, "i" was used to index the month itself (not the collection of values for a given month). I ask because I can imagine another form of "variance" that is more in line with my expectations, but i'm not entirely sure how it should be interpreted with respect to the variance I described above (or if it's even functionally different from what i described above): This is where \overline{ET} is the long term monthly mean of that particular variable (not the annual mean), and where the variance is the variance of the annual realization of that variable about it's long term monthly mean. I think the author's framework addresses the former definition, but am not sure. Is there a significant difference between these two interpretations? If so, what are the different types of questions that you might address with one approach or the other? Additionally, in the case of the first description (average deviation in a given month from a long term annual mean), why is this simply not referred to as seasonality? Presumably this form of "variability" can't be used to address questions relating to long term trends, etc. I apologize if this long-winded question is a bit convoluted; I'm wrestling with some of these concepts for the first time! Thanks for any additional clarification.

COMMENTS:

-It would be helpful if the authors included units when introducing terms; e.g. What is "M" and what are its units?

-Lines 53 - 65: veg change and disturbance?

-Lines 56-57: Why the "but"?

-Lines 67-68: What do the authors mean by "which has been the foundation for decomposing ET or runoff variance and is expressed as:". What has been the foundation for decomposing ET? Are the authors saying that "snowmelt influence has been the foundation for decomposing ET"? I'm not sure what that means.

-Lines 131-132: \Delta S is computed as difference in GLDAS soil moisture down to 2m between months. However, the authors explicitly refer to groundwater as being important with respect to storage change impacts on ET in their introduction. Can these shallow soil moisture measurements reliably represent total storage changes in the catchment? Presumably, in these semi-arid basins, significant storage dynamics occur below 2m depth, both in the deep unsaturated zone and deeper groundwater. What are the consequences of this for the author's findings? Will the impact of storage changes be significantly underestimated?

-Line 138: This seems important. Some overview of the Yang 2009 method would be helpful.

-Line 142: What is F? Assuming it's percent forest cover. It's unclear why we need this, and its relationship to M.

-Line 157: Perhaps useful to point out the parallel to Zeng and Cai, a further elaboration of "effective" precipitation. In their case, this included precip and deltaS. Here, snowmelt is added.

-Line 162: So, ET is obtained as the residual of a mass balance, and then this nonlinear equation is solved for "n" for each value of ET? It seems strange to me not to use the GLDAS-estimated ET (which is available), given that this is how \Delta S is specified.

-Line 170: GLDAS specifically has a snowmelt band. Why not use that, given using it for other aspects of the analysis? Might be more consistent?

-Line 180: The authors state March to July are the major snowmelt months. Why then do the authors then only perform their analysis April to July? Also, why not just apply

the analysis for all months of the year? What is the purpose of leaving out the rest of the year?

-Line 186: "M" has not been sufficiently defined leading up to this section.

-Line 194: Is there any basis (e.g. citation) for this functional dependence? While I agree that vegetation will play a role in determining 'n', it's also true that 'n' likely depends on other catchment features, such as soil water storage capacity. I guess the question, then, is whether these other drivers can be assumed constant through time, and thus somehow justifiably lumped into the fitted constant parameter 'a' in Equation 8. Can the authors confirm that vegetation is likely the only non-static component of the exponent 'n' through a brief review of such mechanistic models? The first one that comes to mind is Porporato et al (2004); though I'm not sure the functional form is identical to Equation 3. Porporato, Amilcare, Edoardo Daly, and Ignacio Rodriguez-Iturbe. "Soil water balance and ecosystem response to climate change." The American Naturalist 164.5 (2004): 625-632.

-Line 216: It's probably obvious to most folks, but the authors should still probably define the overbar as some long term mean. Also, would \bar{ET} also equal the long-term mean ET from Equation 3? Probably best to try to stay consistent with notation if possible.

-Line 227: What is the function "F"? It is not defined. It is referred to as a "factor" in Line 233. Also, what is the underscore notation used here e.g. "R_M". Presumably these correspond to the terms in Equation 14, but that's not very clear, and the notation is not explained or defined.

-Line 240 - 244: This is an unexpected addition that I don't fully understand. Are the authors analyzing results for different representations of effective precipitation? If so, why, and where was this motivated? I don't think it was outlined in the methods. It looks like the authors use 3 different forms of increasing complexity; precip alone; precip plus snowmelt; precip plus snowmelt plus storage differential.

-Figure 3: I'm also a bit confused on this figure. Should it be the case that the points fall on the correspondingly colored curves? Were the curves generated by fitting to the points? Why should a single curve be fit across the ensemble of ET/Pe values for each month? Isn't it reasonable to expect that even the same month in different years will have different values for "n" due to interannual variability in factors that determine "n"? (this relates to my general question about timescales at the start of the review).

-Line 245: Can the authors explain this statement? I don't understand the significance. Is this just to say that if you don't account for all potential fluxes into the rooting zone, the mass balance might be incorrect?

-Line 261: Is it true that \Delta S is expected to be small or zero if there are no inter-annual storage changes?

-Line 304 - 306: I don't think this is an explanation; it's a restatement of the finding that vegetation has a larger impact on ET variance when water is not limiting. The authors still have not answered (or ventured a hypothesis) as to why ET is more sensitive to variability in vegetation cover when water is not a limiting factor? I can think of a couple of vague hypotheses, but would love to see a bit more discussion from the authors on this point; it seems central to the paper.

-Line 313: A downward trend with respect to increasing aridity? It would be helpful if the authors continued to explicitly state the dependent and independent variables when talking about trends.

-Line 318: Elasticity has not been defined up to this point. This is an important concept that the authors should explain more clearly around Equations 13 and 14.

-Line 318: I think it would be very helpful if the authors more explicitly described this idea that the contribution is dependent on both the magnitude of the variance of the driving variable as well as the elasticity.

-Lines 320 - 324: The model developed here cannot speak to these non-stationary

changes though, correct? The analysis here is only pertinent to intra-annual variability attribution, as the variance under consideration is that of the average of the monthly deviation from an annual mean, as opposed to the year-to-year variance of a particular variable about it's long term monthly mean? Again, this relates to my timescale question at the start of the review.

-Line 330: What is a "good" vegetation condition?

-Line 392: "Corrected" I assume should be "correlated"?

---

## Referee Comment (RC2) · Anonymous Referee #2 · 14 Jan 2021

Review of HESS Manuscript MS#hess-2020-535

Title: Attribution of growing season evapotranspiration variability considering snowmelt and vegetation changes in the arid alpine basins

Authors: Ning et al

This manuscript aimed to extend previous framework of temporal variance decomposition in snow-dependent basins by incorporating the effects of snowmelt and vegetation changes. The topic is interesting and the manuscript is well structured. However, I have serious concerns with the methods and results (especially the robustness of the estimates of water cycle components) in the manuscript.

[Figure]

Comments

1. In this study, the total water storage is estimated using the GLDAS soil moisture and plant canopy surface water. Is this estimation reliable? More details about the methods (or additional comparison) may be needed to show the robustness of the total water storage estimation.

2. The degree-day model is used to estimate the equivalent of snowmelt runoff. In this model, the degree-day factors (DDF) in the study basins are fixed (if my understanding is correct here) and vary from 1.7-4.0 mm/day·°C. Is there any uncertainty/validation of these factors? How the variation of the DDF could possibly affect the results of snowmelt runoff?

3. The total water storage and snowmelt runoff estimates are then used to calculate ET. Is the obtained ET reliable in terms of the above two comments?

4. I do not understand the results in Fig. 3. For example, we can see there are black dots in panel (b) (Pe=R-dS) with VERY low ET/Pe values (close to zero). If I understand this correctly, when replace Pe with R+Qm-dS in panel (d), the ET/Pe should decrease as the Qm is positive (Table 1). It means that these low ET/Pe values in panel (b) should be more close to zero (close to x-axis) in panel (d). However, I did not see any black dots close to x-axis. WHERE are they? The results in Fig. 3 are confusing and do not make sense.

5. Do the Qs in the equations and the Qm in the figures have the same physical meaning? If so, please keep the symbols consistent in the manuscript.

6. In this manuscript, the term "temporal variance" is used in growing season by simply extending previous studies (e.g., Liu et al 2019). Is the definition of "temporal variance" in the growing season in this study the same as that in previous work? I cannot understand how it works in math. . .

535, 2020.

HESSD

Interactive
comment

---

## Author Comment (AC1) · 2 Feb 2021

Thank you for the constructive comments. Several figures and equations can not be exhibited normally here, thus a clearer version of our response was submitted as supplement.

The authors extend existing Budyko-type approaches for decomposing monthly ET variance (eg Liu et al 2019) amongst variances (average monthly deviation from an annual mean value) in underlying physical drivers of plant water use (e.g. rainfall). In particular, the model extension now accounts for variance in snowmelt fluxes and variance in vegetation cover. The manuscript is a logical extension of work previously

published on the topic. However, I do have serious concerns with clarity of presentation in some parts of the manuscript, as well as the underlying "consistency" of the datasets used in the study (detailed in specific comments below).

RESPONSE: As for the "consistency" of the datasets, three variables cannot be directly observed at the basin scale, including evapotranspiration (ET), water storage change ($\Delta$S) and snowmelt runoff (Qm), but can be indirectly estimated from different sources of datasets. In this study, $\Delta$S and Qm were estimated by GLDAS data and the degree-day model, respectively. ET was obtained using the water balance equation based on the data of rainfall, runoff, $\Delta$S and Qm. As you mentioned, GLDAS-$\Delta$S may ignore the groundwater storage and lead to errors; however, it seemed that this is the best option. For example, GRACE data is superior for estimating $\Delta$S; however, its coarse spatial resolution may result in even larger errors (please find detailed response in comment 5).

GLDAS has a snowmelt band, but we selected the degree-day model for Qm because of the following aspects. On the one hand, large uncertainties exist in the snow data from GLDAS products. On the other hand, the major input data in the degree-day model we used are measured, which can provide more accurate results of snowmelt runoff. The previous studies using different methods indicated that our modelled Qm is reliable (see comment 10).

Similarly, because of the uncertainties in the forcing data and modelling algorithms of GLDAS-ET, the estimated ET from the water balance equation is more reasonable (see comment 9). To validate the reliability of our ET, we conducted a comparison between our estimated ET, ET_GLDAS and ET from a dataset of "ground truth of land surface evapotranspiration at regional scale in the Heihe River Basin (2012-2016) ETmap Version 1.0", respectively. The results showed that our estimated ET fits better with ETmap compared to GLDAS-ET, suggesting our estimated ET is acceptable.

Of course, in order to keep the "consistency" of the datasets, the abovementioned

three variables from GLDAS can be used in the revised manuscript. However, we think, data reliability is more important than data consistency, especially for those data with large uncertainties. For any employed dataset, uncertainties would exist in the three estimated variables, a new section about the uncertainties will thus be added in the revised manuscript:

**4.5 Uncertainties**

[revised manuscript text omitted]

compared with ETmap.

Figure S4. Comparison of monthly ET derived from water balance equation and ETmap during 2012-2014.

I also have a general question about the overall approach (that will probably reveal my own ignorance about these methods!). When I think about "variability" in ET, I first think about the year-by-year variation in the magnitude of ET in a particular month. However, if I'm understanding this manuscript correctly, the variability that is under consideration is the average of the monthly deviation of ET (or underlying drivers) from a long-term annual average. Is this interpretation correct? That is, in Equation 12 does noverline ET equal the long term annual mean, and does the index "i" in this case index all of the values for a given month? If so, this is somewhat confusing, as in the previous sections, "i" was used to index the month itself (not the collection of values for a given month). I ask because I can imagine another form of "variance" that is more in line with my expectations, but i'm not entirely sure how it should be interpreted with respect to the variance I described above (or if it's even functionally different from what i described above): This is where noverline ET is the long term monthly mean of that particular variable (not the annual mean), and where the variance is the variance of the annual realization of that variable about it's long term monthly mean. I think the author's framework addresses the former definition, but am not sure. Is there a significant difference between these two interpretations? If so, what are the different types of questions that you might address with one approach or the other? Additionally, in the case of the first description (average deviation in a given month from a long term annual mean), why is this simply not referred to as seasonality? Presumably this form of "variability" can't be used to address questions relating to long term trends, etc. I apologize if this long-winded question is a bit convoluted; I'm wrestling with some of these concepts for the first time! Thanks for any additional clarification.

RESPONSE: The unbiased sample variance in equation 12 is estimated by the concept of statistics, not derived by previous studies or us. I would like to clarify the specific

calculation as follows: in this study, with data of growing season (April to September) during 2001-2014, the sample size was 6 months/year×14 years=84 months, i.e. N=84 in equation 12. The calculation regarded all the months as a group or a time series of data, and did not conduct calculation for each calendar month. In consequence, i is used to index time series of month from 1 to N. (ET ) Ìˇis the long-term average of ET for 84 months. As such, one time series of data can only had one variance. It is known that a small test set size leads to a large bias in the estimate of the true variance between design sets (Geng et al., 1979; Wickenberg-Bolin et al., 2006). Comparing with conducting calculation for each calendar month, the calculation by us and other researchers (Liu et al., 2019; Ye et al., 2015; Zeng and Cai, 2015; Zeng and Cai, 2016) can obtain larger sample size. In fact, our variance can also refer to the ET seasonality, as to it reflect the intra-annual change in ET. In the revised version, we will explain the related variables more clearly:

The unbiased sample variance of ET ($\sigma\_ET^2$) is defined as:

$\sigma\_ET^2 = 1/(N-1) \sum\_(i=1)N(ET\_i - (ET))2 = 1/(N-1) \sum\_(i=1)N(\Delta \tilde{a}\breve{A}\acute{U}ET\tilde{a}\breve{A}\mathring{U}\_i)\tilde{a}\breve{A}\mathring{U}^2$
(12)

where (ET) Ìˇ is the long term monthly mean of ET. N is the sample size, it equals 84 in this study (6 months/year×14 years=84 months). i is used to index time series of month from 1 to N.

COMMENTS:

1) It would be helpful if the authors included units when introducing terms; e.g. What is "M" and what are its units?

RESPONSE: The units of related variables will be added in the revised version. M is vegetation coverage and is dimensionless. "M" will be introduced in more details in Line 140:

The monthly normalized difference vegetation index (NDVI) at a spatial resolution of

1km from the MODIS MOD13A3.006 product was used to assess vegetation coverage (M), which can be calculated from the method of Yang et al. (2009):

M=(NDVI-ãĂŰNDVIãĂŮ_min)/(ãĂŰNDVIãĂŮ_max-ãĂŰNDVIãĂŮ_min )

where NDVImax and NDVImin are the NDVI values of dense forest (0.80) and bare soil (0.05).

2) Lines 53 - 65: veg change and disturbance?

RESPONSE: Vegetation change is more suitable. Vegetation change is the final results no matter it is disturbed by any environmental change.

3) Lines 56-57: Why the "but"?

RESPONSE: With the given precipitation, if vegetation condition is improved, the transpiration will increase to lead to higher ET. According to the water balance equation, the increasing ET will result in decreasing runoff, i.e., the ratio of Qr to P. To clarify this, it was replaced with "and" in this version.

4) Lines 67-68: What do the authors mean by "which has been the foundation for decomposing ET or runoff variance and is expressed as:". What has been the foundation for decomposing ET? Are the authors saying that "snowmelt influence has been the foundation for decomposing ET"? I'm not sure what that means.

RESPONSE: The short-term water balance equation was the foundation of decomposing ET/or runoff variance. Its general form can be expressed as: P=ET+Qr+ΔS. But this equation is not suitable for the regions where the hydrology is highly dependent on winter mountain snowpack and spring snowmelt runoff. Thus, the water balance equation should be modified to consider the impacts of snowmelt on runoff in short-term time scale. To clarify this, it will be revised as:

The short-term water balance equation was the foundation of decomposing ET/or runoff variance. Its general form can be expressed as:

[Figure]

P=ET+Qr+ΔS (1)

where P, including liquid (rainfall) and solid (snowfall) precipitation, is the total water source of hydrological cycle. But this equation is not suitable for the regions where the land surface hydrology is highly dependent on winter mountain snowpack and spring snowmelt runoff. It has been reported that annual Qr originating from snowmelt accounted for 20%-70% of the total runoff across the world, including west United States (Huning and AghaKouchak, 2018), coastal areas of Europe (Barnett et al., 2005), west China (Li et al., 2019b), northwest India (Maurya et al., 2018), south of the Hindu Kush (Ragettli et al., 2015), and high-mountain Asia (Qin et al., 2020). In these regions, the mountain snowpack serves as a natural reservoir that storing cold-season P to meet the warm-season water demand (Qin et al., 2020; Stewart, 2009).Thus, the water balance equation should be modified to consider the impacts of snowmelt on runoff in short-term time scale.

5) Lines 131-132: Delta S is computed as difference in GLDAS soil moisture down to 2m between months. However, the authors explicitly refer to groundwater as being important with respect to storage change impacts on ET in their introduction. Can these shallow soil moisture measurements reliably represent total storage changes in the catchment? Presumably, in these semi-arid basins, significant storage dynamics occur below 2m depth, both in the deep unsaturated zone and deeper groundwater. What are the consequences of this for the author's findings? Will the impact of storage changes be significantly underestimated?

RESPONSE: Yes, GLDAS-ΔS ignores the change in groundwater. For groundwater change, the best option is GRACE-ΔS; however, it is not applicable in the study area since the low spatial resolution of GRACE (1°×1°) would lead to large errors in small basins. Instead, GLDAS-ΔS is appropriate in representing the basin-scale water storage change. First, GLDAS has high spatial resolution of 0.25°×0.25°. Second, the groundwater change in west China is small and can be ignored. Specifically, Du et al. (2016) used the abcd model to quantitatively determine monthly variations of water

balance for the sub-basins of Heihe River (including basins 3-5 in our study ) and found that soil water storage change have effects on monthly water balance, whilst the impact of monthly groundwater storage change is negligible. To clarify the uncertainties, a new section will be added in the revised manuscript. The details can be found at the beginning.

6) Line 138: This seems important. Some overview of the Yang 2009 method would be helpful.

RESPONSE: M" will be introduced in more details and the specific revision can be found in comment 1.

7) Line 142: What is F? Assuming it's percent forest cover. It's unclear why we need this, and its relationship to M.

RESPONSE: F is percent forest cover. It was used to explain the finding of "better vegetation condition, especially larger forest cover, could result in stronger impacts on ET variance" in Line 302-306. In your 20th comment, you think that this explanation is just a restatement of the finding. We will thus delete the related text of F in Line 141-144 and Line 304-306, and will give discussion according to your suggestion in the 20th comment.

8) Line 157: Perhaps useful to point out the parallel to Zeng and Cai, a further elaboration of "effective" precipitation. In their case, this included precip and deltaS. Here, snowmelt is added.

RESPONSE: This part will be revised as:

$$ET=(P\_e\times E\_0)/ãĂŰ(ãĂŰP\_eãĂŮ\hat{}n+E\_0\hat{}n)ãĂŮ\hat{}(1/n) \quad (3)$$

where n is the controlling parameter of the Choudhury–Yang equation. Pe is the total available water supply for ET. In previous studies, Pe included P and $\Delta S$ (Pe=P-$\Delta S$) on finer time scale. But snowmelt runoff should also be considered in the snow-dependent basins.

9) Line 162: So, ET is obtained as the residual of a mass balance, and then this nonlinear equation is solved for "n" for each value of ET? It seems strange to me not to use the GLDAS-estimated ET (which is available), given that this is how Delta S is specified.

RESPONSE: Yes, your understanding of the calculation of ET and n is right. When we chose the datasets used in our manuscript, the observed data was first choice. However, it is hard to obtain the observed ET at catchment scale. GLDAS-ET is indeed available. But it has been found that GLDAS products failed to reproduce the water balance-based annual ET time series, which was considered as measured ET, over most basins in China (Bai and Liu, 2018). The errors may come from the uncertainty of its forcing data and model algorithms. On one hand, the precipitation data come from the Princeton Global Fording dataset, which is a reanalysis dataset generated from a climate model. The spatial resolution is only $2° \times 2°$ (Sheffield et al., 2006). The low spatial resolution of forcing data should surely affect ET accuracy, especially in small basins. On the other hand, GLDAS products used Penman-Monteith equation to estimate ET. In this equation, the soil water stress factor is critically important for plant transpiration suppression. However, this factor was implicitly considered by GLDAS products with the vapor pressure deficit (VPD). It is potentially problematic to use VPD to reflect soil water stress for transpiration, especially in drier regions. Some other promising recently released high-resolution ET products, such as GLEAM v3.2 and CLSM v2.0 also have similar problems. In the introduction section, we have illustrated that the short-term water balance equation was the foundation of decomposing ET/or runoff variance (see comment 4). In many present studies of Budyko, water balance equation is also usually used to obtain the ET values (Liu et al., 2018; Wang, 2012a; Yang et al., 2009) . Further, ET obtained from water balance equation was usually considered as "observed ET" and used to validate modeled ET(Bai and Liu, 2018; Liu et al., 2016). The advantage of this method is that the water balance terms except for ET come from the direct observations. In our manuscript, except for observed rainfall and runoff items, the water balance equation also includes snowmelt (Qr) and

ΔS items. Even Qr and ΔS do not directly observed, the key parameters of Qr model is calculated using measured data; further, GLDAS-ΔS is also acceptable, which has been explained in comment 5. To sum up, we think ET obtained from water balance equation should be relatively reasonable compared with global ET products.

To validate the reliability of our ET, we conducted a comparison between our estimated ET, ET_GLDAS and ET from a dataset of "ground truth of land surface evapotranspiration at regional scale in the Heihe River Basin (2012-2016) ETmap Version 1.0", respectively. This ET dataset was published by National Tibetan Plateau Data Center. It was upscaled from 36 eddy covariance flux tower sites (65 site years) to the regional scale with five machine learning algorithms, and then applied to estimate ET for each grid cell (1 km × 1 km) across the Heihe River Basin each day over the period 2012–2016. It has been evaluated to have high accuracy Basins 3,4,5 in our study belongs to the headwater sub-basins of Heihe River, and our monthly ET from April to September during 2012-2014 was thus compared with ETmap (see Figure S4). The results showed that our estimated ET fits better with ETmap compared to GLDAS-ET and basically fell around the 1:1 line. Moreover, ET_GLDAS values is obviously smaller than ETmap. Even in the July and August, monthly ET_GLDAS is less than 60 mm, which is unreasonable in this region. Thus, it can be concluded that our estimated ET by water balance equation is acceptable. The details can be found at the beginning.

figure. Comparison of monthly ET derived from GLDAS product and ETmap during 2012-2014.

10) Line 170: GLDAS specifically has a snowmelt band. Why not use that, given using it for other aspects of the analysis? Might be more consistent?

RESPONSE: The snow model used by GLDAS2-Noah is the Noah land surface model, which has a single-layer snow scheme. And it is forced using the Princeton meteorological forcing dataset. It has been found that there remains substantial uncertainty about the representation of snow on the ground in many reanalysis and GLDAS products, as

evidenced by the wide spread of snow water equivalent and snow depth simulations in such systems (e.g., (Broxton et al., 2016; Mudryk et al., 2015)). For example, GLDAS products underestimate forcing data, including precipitation and snow season temperature, which undeniably contributes to these products having low snow water equivalent. Furthermore, GLDAS products that predict more snow ablation at near-freezing temperatures have larger underestimates of snow water equivalent. In contrast, the major input data in the degree-day model we used are measured. Specifically, the temperature data comes from meteorological stations and the degree-day factor is surveyed by difference GPS for each basin. Even the snow cover data is obtained from remote sensing product, its higher spatial resolution (1km×1km) could reflect slight change of snow cover area and lead to relatively accurate modelling of snowmelt runoff.

We also compared our snowmelt runoff values with other studies. In comparison, the contribution of snow meltwater to runoff (Fs) was 12.9% in Basin 2 during 1971-2015 by using Spatial Processes in Hydrology model(Li et al., 2019), while Fs was 25% in Basin 3 from 2001 to 2012 based on geomorphology-based ecohydrological model (Li et al., 2018), <10% in Basin 6 during 1961-2006 by using SRM model (Gao et al., 2011). Our results indicated that the Fs in Basin 2, 3 and 6 were 14.8%, 24.5% and 6.7%, respectively, which were close to those from different models. It can be concluded that our snowmelt runoff value is acceptable. The details can be found at the beginning.

11) Line 180: The authors state March to July are the major snowmelt months. Why then do the authors then only perform their analysis April to July? Also, why not just apply the analysis for all months of the year? What is the purpose of leaving out the rest of the year?

RESPONSE: In this study, we focused on the ET variability in growing season, which is from April to September in this region. Thus, the beginning of snowmelt months was set as April. One of the major issues we care about is quantifying the contributions of vegetation change on ET variance on finer time scale by developing the relationship between Budyko controlling parameter n and vegetation coverage M on monthly scale.

At first, we explored the relationship between n and M for all months of the year, but their relationship was not significant in most basins. We found that the abnormal points in non-growing season influenced this relationship, which means the impact of vegetation on ET variance in non-growing season is very weak. Thus, in order to decrease the attribution error, we focused on the analysis in growing season.

12) Line 186: "M" has not been sufficiently defined leading up to this section.

RESPONSE: the specific revision of "M"can be found in comment 1.

13) Line 194: Is there any basis (e.g. citation) for this functional dependence? While I agree that vegetation will play a role in determining 'n', it's also true that 'n' likely depends on other catchment features, such as soil water storage capacity. I guess the question, then, is whether these other drivers can be assumed constant through time, and thus somehow justifiably lumped into the fitted constant parameter 'a' in Equation8. Can the authors confirm that vegetation is likely the only non-static component of the exponent 'n' through a brief review of such mechanistic models? The first one that comes to mind is Porporato et al (2004); though I'm not sure the functional form is identical to Equation 3. Porporato, Amilcare, Edoardo Daly, and Ignacio RodriguezIturbe. "Soil water balance and ecosystem response to climate change." The American Naturalist 164.5 (2004): 625-632.

RESPONSE: Except for our previous study in HESS (Ning et al, 2017), Yang et al. (2009) in WRR and Liu et al. (2018) in HESS also used power function to fit the relationship between controlling parameter and vegetation coverage (M). These works will be cited in our revised manuscript.

Except for vegetation condition, other factors, such as soil property, topography and climate seasonality will also influence parameter controlling parameter. But multicollinearity may occur when the explanatory variables are intercorrelated, which will, in turn, induce a series of problems. For example, the effects of individual explanatory variables may not be precisely estimated and the regression coefficients may become

highly unstable (Mjelde et al., 1991). Therefore, it is necessary to check the interactions between explanatory variables and select independent variables when developing expressions of the controlling parameters. This work has been done in our previous study: "Ning, Tingting, Zhou, Sha, et al., 2019. Interaction of vegetation, climate and topography on evapotranspiration modelling at different time scales within the Budyko framework. Agricultural and Forest Meteorology, 275: 59-68". The changes in landform, such as topography or soils, are gentle and can be ignored. Therefore, we focused on the interactions among vegetation coverage(M), climate seasonality index (SAI). We found that, on annual scale, M and SAI were significantly related to controlling parameter, while being independent from each other; in consequence, both of them should be parameterized into the Budyko model. In this study, we only considered M while ignored SAI when parameterizing n because the covariance item between rainfall and potential evapotranspiration, i.e. $\varepsilon 1\varepsilon 4cov(R, E0)$, in equation (13) can represent the climate seasonality. According to your suggestion, a brief review about the impact factors of Budyko controlling parameter will be added in the discussion of revised manuscript. The details can be found at the beginning of this response.

14) Line 216: It's probably obvious to most folks, but the authors should still probably define the overbar as some long term mean. Also, would nbar{ET also equal the longterm mean ET from Equation 3? Probably best to try to stay consistent with notation if possible.

RESPONSE: (ET) ÌĚ will be defined as "long-term mean of monthly ET" in the revised manuscript. Initially, the Choudhury-Yang equation, i.e. equation (3), was derived on long-term scale, thus ET in this equation represented ET on the long-term scale, and it will be revised as (ET) ÌĚ in the revised manuscript.

15) Line 227: What is the function "F"? It is not defined. It is referred to as a "factor" in Line 233. Also, what is the underscore notation used here e.g. "R_M". Presumably these correspond to the terms in Equation 14, but that's not very clear, and the notation is not explained or defined.

[Figure]

RESPONSE: Sorry for our carelessness. In the revised manuscript, "F" will be defined as "the individual contributions of each factor"; each two factors linked by underscore represent the interaction effects between them.

16) Line 240 - 244: This is an unexpected addition that I don't fully understand. Are the authors analyzing results for different representations of effective precipitation? If so, why, and where was this motivated? I don't think it was outlined in the methods. It looks like the authors use 3 different forms of increasing complexity; precip alone; precip plus snowmelt; precip plus snowmelt plus storage differential.

RESPONSE: Yes, Figure 3 presents the water balance in the monthly scale of all the study basins in the Budyko's framework with four different computations of aridity index or ET ratio: i. $ET=R-Qr$ when R is considered as water supply ($Pe=R$); ii. $ET=R-\Delta S-Qr$ when $R-\Delta S$ is considered as water supply ($Pe= R-\Delta S$); iii. $ET=R+Qm-Qr$ when $R+Qm$ is considered as water supply ($Pe= R+Qm$); iv. $ET=R-\Delta S +Qm-Qr$ when $R-\Delta S+Qm$ is considered as water supply ($Pe= R-\Delta S+Qm$).

The motivation is: the Budyko framework was originally derived on long-term scale. Then it was gradually extended to characterize and predict the interannual variability of ET and the runoff fluxes on short time scales (including interanual and monthly scales). Some studies also showed that the Budyko framework was not suitable to represent ET variation on short time scales, because of the data points drew by ET ratio and dryness index beyond the two limit curves of Budyko framework (Chen et al., 2013; Du et al., 2016; Wang, 2012b). These studies found that ignoring $\Delta S$ is the main reason (see Figure 11 by (Du et al., 2016); Figure 3 by (Chen et al., 2013)). Thus, validating the feasibility of using Budyko equation for variability of ET on the short time scale is the foundation.

Considering different combinations of water supply to ET is the main method for validation. In this study, except for $\Delta S$, snowmelt runoff (Qm) is an important item of monthly water balance equation. Four combinations of water supply were thus assumed to

prove the importance of considering ∆S and Qm into Budyko framework on monthly scale in the original manuscript. In this version, to avoid confusion, we only considered three combinations of water supply, i.e., Pe=R, Pe= R-∆S and Pe=R-∆S+Qm.

The motivations will be added and the related expression will be revised. You pointed out that the operation of this part was not outlined in the methods. As they are used to show the distribution of data points of E0/Pe and ET/Pe under the Budyko framework, the figure is enough even without description in the methods.

4.1 The effects of monthly storage change and snowmelt runoff in the Budyko framework

The Budyko framework is usually used for analyses of long-term average catchment water balance; however, it was employed for the interpretation of the monthly variability of the water balance in this study. Thus, it's very necessary to validate the feasibility of Budyko equation for monthly variability. Furthermore, the impact of ∆S on the representation of Budyko framework on finer time scale has assessed by several studies (Chen et al., 2013; Du et al., 2016; Liu et al., 2019; Zeng and Cai, 2015). However, the impact of Qm and its combined effects with ∆S in snowmelt-dependent basins are mostly ignored. Therefore, we present the water balance in the monthly scale of six basins in the Budyko's framework with three different computations of aridity index (ÏȚ=E0/Pe) or ET ratio (ET/Pe) in Figure 3. In Figure 3a, ET=R-Qr when R is considered as water supply, i.e., Pe=R. The points of monthly ET ratio and aridity index in April and May were well below Budyko curves in 6 basins; monthly ET ratio was even negative in several year, which means the local rain are not the only sources of ET in this area, especially in spring. In Figure 3b, ET=R-∆S-Qr with Pe= R-∆S. Compared with figure 3a, the way-off points in April and May were improved to a certain extent but negative points still existed, suggesting that except for R, ∆S also play a significant role in maintaining spring ET, but the variability of ET cannot be completely explained by these two variables. In Figure 3c, ET=R-∆S+Qm-Qr with Pe=R-∆S+Qm. Compared to the points in Figures 3a-b, all points focused on Budyko's curves more closely in each

basin when Pe=R+Qm-ΔS (Figure 3c). From this comparison, it can be concluded that the Budyko framework is applicable to the monthly scale in snowmelt-dependent basins, if the water supply is described accurately by considering ΔS and Qm.

Figure 3 Plots for aridity index vs. evapotranspiration index scaled by available water supply for monthly series in growing season. Total water availability is (a) R, (b) R-ΔS and (c) R-ΔS +Qm. The n value for each Budyko curve is fitted by long-term averaged monthly data.

17) Figure 3: I'm also a bit confused on this figure. Should it be the case that the points fall on the correspondingly colored curves? Were the curves generated by fitting to the points? Why should a single curve be fit across the ensemble of ET/Pe values for each month? Isn't it reasonable to expect that even the same month in different years will have different values for "n" due to interannual variability in factors that determine "n"? (this relates to my general question about timescales at the start of the review).

RESPONSE: Theoretically, the points should fall on the Budyko curves. But deviations from the Budyko curve have been detected in many previous studies. In addition to climate conditions, other variables including vegetation, soil, topography and climate seasonality, also influence the variability of regional water balances (Wang, 2012b; Yang et al., 2007). All these factors can be encoded into the controlling parameter of the Budyko equations. In this study, the vegetation coverage was chosen to explain the monthly variability of n, which was obtained by equation 5 for each month. To make the figure clearer, the mean annual n for each month was used to draw the Budyko curve. This operation was also adopted by many previous studies (Du et al., 2016; Liu et al., 2018; Ning et al., 2017).

18) Line 245: Can the authors explain this statement? I don't understand the significance. Is this just to say that if you don't account for all potential fluxes into the rooting zone, the mass balance might be incorrect?

RESPONSE: Yes. In order to obtain the right ET values, the mass balance should
consider all potential fluxes. If this is not done, the abnormal data point will be observed in Figure 3, such as the negative monthly ET ratio in Figure 3a-b.

19) Line 261: Is it true that nDelta S is expected to be small or zero if there are no interannual storage changes?

RESPONSE: We checked the original data and recalculated the mean and the standard deviation values of $\Delta S$ and confirmed this result is correct. Here, what we found is that the intra-annual changes of water storage is relatively large, but its mean monthly value was small. This is because $\Delta S$ in some months is positive but in some months is negative.

20) Line 304 - 306: I don't think this is an explanation; it's a restatement of the finding that vegetation has a larger impact on ET variance when water is not limiting. The authors still have not answered (or ventured a hypothesis) as to why ET is more sensitive to variability in vegetation cover when water is not a limiting factor? I can think of a couple of vague hypotheses, but would love to see a bit more discussion from the authors on this point; it seems central to the paper.

RESPONSE: We agreed with you. The related text of F will be deleted. We will give discussion according to your suggestion in the revised manuscript:

C(M) showed an increasing trend from 0.5% to 9.5% with the decreasing ÏŢ, implying that the contribution of vegetation change to ET variance was larger in relatively humid basin. It can be explained that transpiration is more sensitive to vegetation change, and thus the higher vegetation coverage could increase the proportion of transpiration to ET in humid regions (Niu et al., 2019; Zhang et al., 2020). The Budyko hypothesis stated that change in ET is controlled by change in available energy when water supply is not a limiting factor under humid conditions (Budyko, 1974; Yang et al., 2006). The increasing M results in the reallocation of available energy between canopy and soil. Specifically, more energy is consumed by canopy thus increases transpiration. Further, Previous studies have found that ET differs greatly among species, because of the difference in

canopy roughness, the timing of physiological functioning, water holding capacity of the soil and rooting depth of the vegetation (Baldocchi et al., 2004; Bruemmer et al., 2012). Generally, forest had larger ET than grassland (Ma et al., 2020; Zha et al., 2010). The fraction of forest area is relatively high and thus lead to the higher contributions to ET for whole basin in the humid region. For example, Wei et al. (2018) showed that the global average variation in the annual Qr due to the vegetation cover change was $30.7 \pm 22.5\%$ in forest-dominated regions on long-term scales, which was higher than our results because of their higher forest cover.

21) Line 313: A downward trend with respect to increasing aridity? It would be helpful if the authors continued to explicitly state the dependent and independent variables when talking about trends.

RESPONSE: This part will be revised as:

Similar as C(R), C(Qm) showed a downward trend with the decreasing Ï̧, ranging from 2.9% to 0.4%.

22) Line 318: Elasticity has not been defined up to this point. This is an important concept that the authors should explain more clearly around Equations 13 and 14.

RESPONSE: $\varepsilon$ in equations 13 and 14 is the partial differentia coefficients of ET to each variable, not the elasticity coefficients. We will define it in equation (11) where it firstly appears.

23) Line 318: I think it would be very helpful if the authors more explicitly described this idea that the contribution is dependent on both the magnitude of the variance of the driving variable as well as the elasticity.

RESPONSE: This part will be revised as:

It can be explained that the contribution of each variable to $\sigma\_ET^2$ was not only the product of the partial differential coefficients, but also relied on its variance value according to equation 13. Specifically, the partial differential coefficients of 0.1 for a

variable means that a 10% change in that variable may result in a change in ET by 1%, which can only reflect the theoretical contribution of each variable. By multiplying the variance value, the actual contribution of each variable could be obtained.

24) Lines 320 - 324: The model developed here cannot speak to these non-stationary changes though, correct? The analysis here is only pertinent to intra-annual variability attribution, as the variance under consideration is that of the average of the monthly deviation from an annual mean, as opposed to the year-to-year variance of a particular variable about it's long term monthly mean? Again, this relates to my timescale question at the start of the review.

RESPONSE: The analysis of this study is only pertinent to intra-annual variability attribution. But it can be used to represent nonstationary changes, but just limited to intra-annual scale. Specifically, the intra-annual variability of ET is related to the intra-annual variability of related factor. Here, we emphasized that the climate warming shifted the timing of snowmelt earlier in the spring in the Qilian Mountains, which resulted in increased soil moisture and a greater proportion of Qm to ET. The shifting of timing of snowmelt earlier in the spring referred to the intra-annual variability of snowmelt period. Thus, we thought it is reasonable.

25) Line 330: What is a "good" vegetation condition?

RESPONSE: This expression is indeed improper. Thus "good vegetation condition" will be revised as "higher vegetation coverage".

26) Line 392: "Corrected" I assume should be "correlated"?

RESPONSE: We are so sorry for our carelessness. "correlated" is right.

* * *
[Figure]

**Fig. 1.** Figure S4. Comparison of monthly ET derived from water balance equation and ETmap during 2012-2014.

[Figure]

**Fig. 2.** Comparison of monthly ET derived from GLDAS product and ETmap during 2012-2014.

[Figure]

**Fig. 3.** Figure 3 Plots for aridity index vs. evapotranspiration index scaled by available water supply for monthly series in growing season. Total water availability is (a) R, (b) R-$\Delta$S and (c) R-$\Delta$S +Qm.

---

## Author Comment (AC2) · 2 Feb 2021

We appreciate your constructive comments. Several figures and equations can not be exhibited normally here, thus a clearer version of our response was submitted as supplement.

Point-to-point responses:

This manuscript aimed to extend previous framework of temporal variance decomposition in snow-dependent basins by incorporating the effects of snowmelt and vegetation changes. The topic is interesting and the manuscript is well structured. However, I

have serious concerns with the methods and results (especially the robustness of the estimates of water cycle components) in the manuscript.

RESPONSE: We appreciate your constructive comments, and will revise the manuscript accordingly. In particular, to validate the reliability of the estimated water cycle components, we compared our results with previous studies, and found that our results are acceptable. Furthermore, the uncertainties were discussed:

4.5 Uncertainties

[revised manuscript text omitted]

River, and our monthly ET from April to September during 2012-2014 in these three basins was thus compared with ETmap.

Figure S4. Comparison of monthly ET derived from water balance equation and ETmap during 2012-2014.

Comments

1. In this study, the total water storage is estimated using the GLDAS soil moisture and plant canopy surface water. Is this estimation reliable? More details about the methods (or additional comparison) may be needed to show the robustness of the total water storage estimation.

RESPONSE: The other reviewer also doubted the reliability of GLDAS-$\Delta$S because it ignored the change in the deeper groundwater. For groundwater change, the best option is GRACE-$\Delta$S; however, it is not applicable in the study area since the low spatial resolution of GRACE ($1°\times1°$) would lead to large errors in small basins. Instead, GLDAS-$\Delta$S is appropriate in representing the basin-scale water storage change. First, GLDAS has high spatial resolution of $0.25°\times0.25°$. Second, the groundwater change in west China is small and can be ignored. Specifically, Du et al. (2016) used the abcd model to quantitatively determine monthly variations of water balance for the sub-basins of Heihe River (including basins 3-5 in our study ) and found that soil water storage change have effects on monthly water balance, whilst the impact of monthly groundwater storage change is negligible. To clarify the uncertainties, a new section will be added in the revised manuscript. The details can be found at the beginning.

2. The degree-day model is used to estimate the equivalent of snowmelt runoff. In this model, the degree-day factors (DDF) in the study basins are fixed (if my understanding is correct here) and vary from 1.7-4.0 mm/dayﾊﾉﾉﾘuC. Is there any uncertainty/validation of these factors? How the variation of the DDF could possibly affect the results of snowmelt runoff?
RESPONSE: Yes, your understanding is correct. It has been found that any changes in climate conditions and the underlying basin characteristics will affect the relative contributions of the heat balance components and cause temporal variations of the DDF (Kuusisto, 1980; Ohmura, 2001). But previous study indicated that there is no significant seasonal change in DDF in Western China (Zhang et al., 2006), which contains our study area. Thus, using the fixed DDF values to estimate snowmelt runoff is acceptable in this area. We also compared our snowmelt runoff values with other studies. In comparison, the contribution of snow meltwater to runoff (Fs) was 12.9% in Basin 2 during 1971-2015 by using Spatial Processes in Hydrology model(Li et al., 2019), while Fs was 25% in Basin 3 from 2001 to 2012 based on geomorphology-based ecohydrological model (Li et al., 2018), <10% in Basin 6 during 1961-2006 by using SRM model (Gao et al., 2011). Our results indicated that the Fs in Basin 2, 3 and 6 were 14.8%, 24.5% and 6.7%, respectively, which were close to those from different models. It can be concluded that our snowmelt runoff value is acceptable. Further, the uncertainties induced by the variation of the DDF will also discussed in revised manuscript. The details can be found at the beginning.

3. The total water storage and snowmelt runoff estimates are then used to calculate ET. Is the obtained ET reliable in terms of the above two comments?

RESPONSE: Even though there are many global ET products, but they have large uncertainty of its forcing data and model algorithms. Taking GLDAS-ET as an example, the precipitation data used come from the Princeton Global Fording dataset, which is a reanalysis dataset generated from a climate model. The spatial resolution is only $2° \times 2°$ (Sheffield et al., 2006). The low spatial resolution of forcing data should surely affect ET accuracy, especially in small basins. On the other hand, GLDAS ET products used Penman-Monteith equation to estimate ET. In this equation, the soil water stress factor is critically important for plant transpiration suppression. However, this factor was implicitly considered by GLDAS products with the vapor pressure deficit (VPD). It is potentially problematic to use VPD to reflect soil water stress for transpiration, especially in drier regions. Some other promising recently released high-resolution ET products, such as GLEAM v3.2 and CLSM v2.0 also have similar problems. Thus, To validate the reliability of our ET, we conducted a comparison between our estimated ET, ET_GLDAS and ET from a dataset of "ground truth of land surface evapotranspiration at regional scale in the Heihe River Basin (2012-2016) ETmap Version 1.0", respectively. This ET dataset was published by National Tibetan Plateau Data Center. It was upscaled from 36 eddy covariance flux tower sites (65 site years) to the regional scale with five machine learning algorithms, and then applied to estimate ET for each grid cell (1 km × 1 km) across the Heihe River Basin each day over the period 2012–2016. It has been evaluated to have high accuracy. Basins 3,4,5 in our study belongs to the headwater sub-basins of Heihe River, and our monthly ET from April to September during 2012-2014 in these three basins was thus compared with ETmap (see Figure S4). The results showed that our estimated ET fits better with ETmap compared to GLDAS-ET and basically fell around the 1:1 line. Moreover, ET_GLDAS values is obviously smaller than ETmap. Even in the July and August, monthly ET_GLDAS is less than 60 mm, which is unreasonable in this region. Thus, it can be concluded that our estimated ET by water balance equation is acceptable. The details can be found at the beginning.

Figure. Comparison of monthly ET derived from GLDAS product and ETmap during 2012-2014.

4. I do not understand the results in Fig. 3. For example, we can see there are black dots in panel (b) (Pe=R-dS) with VERY low ET/Pe values (close to zero). If I understand this correctly, when replace Pe with R+Qm-dS in panel (d), the ET/Pe should decrease as the Qm is positive (Table 1). It means that these low ET/Pe values in panel (b) should be more close to zero (close to x-axis) in panel (d). However, I did not see any black dots close to x-axis. WHERE are they? The results in Fig. 3 are confusing and do not make sense.

RESPONSE: Yes, as the Qm is positive, Pe with R+Qm-dS should increase compared

with Pe with R-dS. But ET also increased, because it equals Pe-Qr according to equation (2) and (4). Thus, E0/Pe in x-axis will decrease while ET/Pe in y-axis will increase when considered Qm in panel d. Furthermore, the larger Qm, the larger increase in ET/Pe while decrease in E0/Pe, which means the black dots will move to the upper left after considering Qm.

The other reviewer also doubted the motivation of this part. Perhaps the simplest revised method is deleting the related content. But we thought it should be reserved as the following reasons: the Budyko framework was originally derived on long-term scale. Then it was gradually extended to characterize and predict the interannual variability of ET and the runoff fluxes on short time scale (including interanual and monthly scales). Some studies also showed that the Budyko framework was not suitable for exhibit ET variation on short time scale, because of the data points drew by ET ratio and dryness index beyond the two limit curves of Budyko framework (Chen et al., 2013; Du et al., 2016; Wang, 2012). These studies found that ignoring $\Delta$S is the main reason (see Figure 11 by (Du et al., 2016); Figure 3 by (Chen et al., 2013)). Thus, validating the feasibility of using Budyko equation for variability of ET on the short time scale is the foundation.

Considering different combinations of water supply to ET is the main method for validation. In this study, except for $\Delta$S, snowmelt runoff (Qm) is an important item of monthly water balance equation. Four combinations of water supply were thus assumed to prove the importance of considering $\Delta$S and Qm into Budyko framework on monthly scale in the original manuscript. In this version, to avoid confusion, we only considered three combinations of water supply, i.e., Pe=R, Pe= R-$\Delta$S and Pe=R-$\Delta$S+Qm.

Further, the related expression will also be revised:

4.1 The effects of monthly storage change and snowmelt runoff in the Budyko framework

The Budyko framework is usually used for analyses of long-term average catchment

water balance; however, it was employed for the interpretation of the monthly variability of the water balance in this study. Thus, it's very necessary to validate the feasibility of Budyko equation for monthly variability. Furthermore, the impact of $\Delta S$ on the representation of Budyko framework on finer time scale has assessed by several studies (Chen et al., 2013; Du et al., 2016; Liu et al., 2019; Zeng and Cai, 2015). However, the impact of Qm and its combined effects with $\Delta S$ in snowmelt-dependent basins are mostly ignored. Therefore, we present the water balance in the monthly scale of six basins in the Budyko's framework with three different computations of aridity index ($\ddot{I}_T$=E0/Pe) or ET ratio (ET/Pe) in Figure 3. In Figure 3a, ET=R-Qr when R is considered as water supply, i.e., Pe=R. The points of monthly ET ratio and aridity index in April and May were well below Budyko curves in 6 basins; monthly ET ratio was even negative in several year, which means the local rain are not the only sources of ET in this area, especially in spring. In Figure 3b, ET=R-$\Delta S$-Qr with Pe= R-$\Delta S$. Compared with figure 3a, the way-off points in April and May were improved to a certain extent but negative points still existed, suggesting that except for R, $\Delta S$ also play a significant role in maintaining spring ET, but the variability of ET cannot be completely explained by these two variables. In Figure 3c, ET=R-$\Delta S$+Qm-Qr with Pe=R-$\Delta S$+Qm. Compared to the points in Figures 3a-b, all points focused on Budyko's curves more closely in each basin when Pe=R+Qm-$\Delta S$ (Figure 3c). From this comparison, it can be concluded that the Budyko framework is applicable to the monthly scale in snowmelt-dependent basins, if the water supply is described accurately by considering $\Delta S$ and Qm.

Figure 3 Plots for aridity index vs. evapotranspiration index scaled by available water supply for monthly series in growing season. Total water availability is (a) R, (b) R-$\Delta S$ and (c) R-$\Delta S$ +Qm. The n value for each Budyko curve is fitted by long-term averaged monthly data.

5. Do the Qs in the equations and the Qm in the figures have the same physical meaning? If so, please keep the symbols consistent in the manuscript.

RESPONSE: We are so sorry for our carelessness. These two symbols all represented

the snowmelt runoff. In the revised version, Qs will be revised as Qm.

6. In this manuscript, the term "temporal variance" is used in growing season by simply extending previous studies (e.g., Liu et al 2019). Is the definition of "temporal variance" in the growing season in this study the same as that in previous work? I cannot understand how it works in math.

RESPONSE: The definition of "temporal variance" in the growing season in this study is same as that in previous work. The only difference is the calculation of sample size (N) in equation 12. In previous studies, they focused on ET or runoff variance for all months. Thus, the sample size was 12 months/year×n years. In this study, we concerned ET variance in the growing season (April to September). Thus the sample size was 6 months/year×n years. The unbiased sample variance in equation 12 is estimated by the concept of statistics, not derived by previous studies or us. I would like to clarify the specific calculation as follows: in this study, with data of growing season (April to September) during 2001-2014, the sample size was 6 months/year×14 years=84 months, i.e. N=84 in equation 12. The calculation regarded all the months as a group or a time series of data, and did not conduct calculation for each calendar month. In consequence, i is used to index time series of month from 1 to N. (ET ) ÌĔis the long-term average of ET for 84 months. As such, one time series of data can only had one variance. It is known that a small test set size leads to a large bias in the estimate of the true variance between design sets (Geng et al., 1979; Wickenberg-Bolin et al., 2006). Comparing with conducting calculation for each calendar month, the calculation by us and other researchers (Liu et al., 2019; Ye et al., 2015; Zeng and Cai, 2015; Zeng and Cai, 2016) can obtain larger sample size. In the revised version, we will explain the related variables more clearly:

The unbiased sample variance of ET ($\sigma\_ET^2$) is defined as:

$$\sigma\_ET^2 = 1/(N-1) \sum\_(i=1)^N (ET\_i - (ET))^2 = 1/(N-1) \sum\_(i=1)^N (\Delta \text{ãĂŰET ãĂŮ}\_i) \text{ãĂŮ}^2$$
(12)

where (ET) ÌĚ is the long term monthly mean of ET. N is the sample size, it equals 84 in this study (6 months/year×14 years=84 months). i is used to index time series of month from 1 to N.

―――――――――――――――――――

[Figure]

**Fig. 1.** Figure S4. Comparison of monthly ET derived from water balance equation and ETmap during 2012-2014.

[Figure]

**Fig. 2.** Comparison of monthly ET derived from GLDAS product and ETmap during 2012-2014.

[Figure]

**Fig. 3.** Figure 3 Plots for aridity index vs. evapotranspiration index scaled by available water supply for monthly series in growing season. Total water availability is (a) R, (b) R-$\Delta$S and (c) R-$\Delta$S +Qm.

---

## Referee Report (RR1)

**Review of HESS Manuscript MS#hess-2020-535: Second review**

Title:        Attribution of growing season evapotranspiration variability considering
              snowmelt and vegetation changes in the arid alpine basins
Authors:      Ning et al

The revised manuscript has greatly improved with some of my concerns addressed especially about the data reliability. However, one very important issue still need be resolved before the manuscript can be published.

As the authors explained in the response, the term "temporal variance" in this study was defined as the ET variance in the growing season (April to September), i.e., the unbiased sample variance of ET in Eqn 13. The sample size was 6 months/year×14 years=84 months, and ET mean in Eqn 13 was calculated as the long-term average for 84 months.

Then what is the physical meaning of that defined "unbiased sample variance of ET"? It is obviously different from the definition of "temporal variance" from previous work (e.g., Zeng and Cai, 2015), and should not be seen as a simple extension of those work. The authors should carefully think about the "temporal variance" definition in this study and provide its physical explanation.

In addition, the ET mean in the "temporal variance" definition in previous studies was the long-term average of all months and kind of fixed (in certain years). In this study, six months (April to September) was selected to define the "temporal variance" and calculate the ET mean. Is it possible that the results could change a lot with different months (e.g., April to July) since the ET mean varies in different months? How the potential divergence of results using different months could be explained?

Another minor comment about the new Fig. S4. We can see that the estimated ET was generally underestimated compared with ETmap. Is it possible to discuss the reason of underestimation and how it could influence the results? Further, why 15 dots in Fig. S4? It should be 18 dots if I understand it correctly?

---

## Author Response (AR2)

The revised manuscript has greatly improved with some of my concerns addressed especially about the data reliability. However, one very important issue still need be resolved before the manuscript can be published.

(1) As the authors explained in the response, the term "temporal variance" in this study was defined as the ET variance in the growing season (April to September), i.e., the unbiased sample variance of ET in Eqn 13. The sample size was 6 months/year×14 years=84 months, and ET mean in Eqn 13 was calculated as the long-term average for 84 months.

Then what is the physical meaning of that defined "unbiased sample variance of ET"? It is obviously different from the definition of "temporal variance" from previous work (e.g.,Zeng and Cai, 2015), and should not be seen as a simple extension of those work. The authors should carefully think about the "temporal variance" definition in this study and provide its physical explanation.

**RESPONSE:** Thanks for your carefully comments very much. Actually, "temporal variance" was also expressed as "unbiased sample variance" in previous work (Liu et al., 2019; Zeng and Cai, 2015), and the specific formula was shown in Eq.13 in Zeng and Cai (2015), and in Eq.6 in Liu et al. (2019). The difference between these two studies is the calculation process of $\Delta$ in this equation. The method of Zeng and Cai (2015) was adopted by the most of previous works (Wu et al., 2017; Ye et al., 2015; Zhang et al., 2016). But we made an extension to Liu et al. (2019) by considering the effects of snowmelt and vegetation changes, because of their calculation process of $\Delta$ is simpler. The "unbiased sample variance" is the concept in probability theory and statistics, and is the expectation of the squared deviation of a random variable from its mean. In other words, it measures dispersion of a set of numbers from their average. As you suggested, we further explain its physical meaning in Line 229-238:

*In this study, the temporal variance of ET reflects the fluctuation of monthly ET in growing season for years, which can be quantified by the unbiased sample variance ($\sigma_{ET}^2$) :*

$$\sigma_{ET}^2 = \frac{1}{N-1}\sum_{i=1}^{N}(ET_i - \overline{ET})^2 = \frac{1}{N-1}\sum_{i=1}^{N}(\Delta ET_i)^2. \qquad (13)$$

*where $\overline{ET}$ is the long term monthly mean of ET. N is the sample size, it equals 84 in this study (6 months/year×14 years=84 months). i is used to index time series of month from 1 to N. $\sigma_{ET}^2$ indicates how far a set of monthly ET in growing season is spread out from their average value. The larger $\sigma_{ET}^2$, the larger fluctuation of ET, and vice versa.*

(2) In addition, the ET mean in the "temporal variance" definition in previous studies was the long-term average of all months and kind of fixed (in certain years). In this study, six months (April to September) was selected to define the "temporal variance" and calculate the ET mean. Is it possible that the results could change a lot with different months (e.g., April to July) since the ET mean varies in different months? How the

potential divergence of results using different months could be explained?

**RESPONSE:** Yes, the results could change a lot using different months. The potential divergence of results using different months could be explained by the different time series of ET, which not only determines ET mean, but also impacts sample size in Eq.13. But the choice of months should have scientific basis. We focused on the ET variance and its attribution in growing season in this study. It has been showed that the growing season is from April to September in previous studies (Jiao et al., 2016; Tian et al., 2013; Xing et al., 2017; Zeng et al., 2019), thus six months (April to September) were selected.

(3) Another minor comment about the new Fig. S4. We can see that the estimated ET was generally underestimated compared with ETmap. Is it possible to discuss the reason of underestimation and how it could influence the results? Further, why 15 dots in Fig. S4? It should be 18 dots if I understand it correctly?

**RESPONSE:** The reason for the underestimation of ET and possible influence were added in section "Uncertainties". Line 445:

*To validate the reliability of our estimated ET, the comparison with $ET_{map}$ from May to September during 2012-2014 was conducted (Figure S4). The results showed that our estimated ET fitted well with $ET_{map}$ and basically fell around the 1:1 line, indicating ET estimated using water balance equation by considering the items of $\Delta S$ and $Q_m$ is acceptable.* **However, it cannot be ignored that our estimated ET was generally lower than $ET_{map}$. The error of rainfall spatial interpolation may explain the underestimation of ET. Most meteorological stations are located at low elevations or in river valleys, but some stations are distributed in high elevations in Qilian Mountain (Figure 1). It has been found that rainfall in mountainous regions is generally larger than that in plain regions (Qiang et al., 2015). Even the topography effect was considered for interpolation, it still resulted in bias in areal rainfall. The best method to improve the quality of spatial rainfall estimation is to increase the density of the monitoring network. However, this process is limited by harsh environment and funds (Buytaert et al., 2006). The error of rainfall will be transferred to contribution quantification of ET variance by underestimating rainfall contribution, while overestimating Qm and $\Delta S$ contribution.**

As for the number of dots, it should be 15 dots. The period of "ETmap" data is from May to September during 2012–2016, thus there are 15 dots in Fig.S4. The "April to September" has been corrected as "May to September" in Line 150, 152 and 442, please check.

References:
Buytaert, W., Celleri, R., Willems, P., Bièvre, B.D. and Wyseure, G., 2006. Spatial and temporal rainfall variability in mountainous areas: A case study from the south Ecuadorian Andes. *Journal of Hydrology*. https://doi.org/10.1016/j.jhydrol.2006.02.031.
Jiao, L., Jiang, Y., Wang, M.C., Kang, X.Y., Zhang, W.T., Zhang, L.N. and Zhao, S.D., 2016. Responses to climate change in radial growth of Picea schrenkiana along elevations of the eastern

Tianshan Mountains, northwest China. *Dendrochronologia*. 40, 117-127. https://doi.org/10.1016/j.dendro.2016.09.002.

Liu, J., Zhang, Q., Feng, S., Gu, X., Singh, V.P. and Sun, P., 2019. Global Attribution of Runoff Variance Across Multiple Timescales. *Journal of Geophysical Research-Atmospheres*. 124(24), 13962-13974. https://doi.org/10.1029/2019jd030539.

Qiang, F., Zhang, M.J., Wang, S., Liu, Y., Ren, Z. and Zhu, X., 2015. Changes of areal precipitation based on gridded dataset in Qilian Mountains during 1961-2012 (In Chinese). *Acta Geographica Sinica*

Tian, J., Su, H.B., Sun, X.M., Chen, S.H., He, H.L. and Zhao, L.J., 2013. Impact of the Spatial Domain Size on the Performance of the T-s-VI Triangle Method in Terrestrial Evapotranspiration Estimation. *Remote Sensing*. 5(4), 1998-2013. https://doi.org/10.3390/rs5041998.

Xing, Q., Wu, B.F., Yan, N.N., Yu, M.Z. and Zhu, W.W., 2017. Evaluating the Relationship between Field Aerodynamic Roughness and the MODIS BRDF, NDVI, and Wind Speed over Grassland. *Atmosphere*. 8(1). https://doi.org/10.3390/atmos8010016.

Zeng, R. and Cai, X., 2015. Assessing the temporal variance of evapotranspiration considering climate and catchment storage factors. *Advances in Water Resources*. 79, 51-60. https://doi.org/10.1016/j.advwatres.2015.02.008.

Zeng, X.M., Evans, M.N., Liu, X.H., Wang, W.Z., Xu, G.B., Wu, G.J. and Zhang, L.N., 2019. Spatial patterns of precipitation-induced moisture availability and their effects on the divergence of conifer stem growth in the western and eastern parts of China's semi-arid region. *Forest Ecology and Management*. 451. https://doi.org/10.1016/j.foreco.2019.117524.